



# Extension of the AIOMFAC model by iodine and carbonate species: applications for aerosol acidity and cloud droplet activation

Hang Yin[1], Jing Dou[2], Liviana Klein[2], Ulrich K. Krieger[2], Alison Bain[3], Brandon J. Wallace[3], Thomas C. Preston[1,3], and Andreas Zuend[1]

[1]Department of Atmospheric and Oceanic Sciences, McGill University, Montreal, Quebec, Canada
[2]Institute for Atmospheric and Climate Science, ETH Zurich, Switzerland
[3]Department of Chemistry, McGill University, Montreal, Quebec, Canada

**Correspondence:** Andreas Zuend (andreas.zuend@mcgill.ca)

**Abstract.** Iodine and carbonate species are important components in marine and dust aerosols, respectively. The non-ideal interactions between these species and other inorganic and organic compounds within aqueous particle phases affect hygroscopicity, acidity, and gas–particle partitioning of semivolatile components. In this work, we present an extended version of the Aerosol Inorganic–Organic Mixtures Functional groups Activity Coefficients (AIOMFAC) model by incorporating the ions $I^-$,

$IO_3^-$, $HCO_3^-$, $CO_3^{2-}$, $OH^-$, and $CO_{2(aq)}$ as new species. First, AIOMFAC ion interaction parameters for aqueous solutions were determined based on available thermodynamic data, such as water activity, mean molal activity coefficients, solubility, and vapor–liquid equilibrium measurements. Second, the interaction parameters for the new ions and various organic functional groups were optimized based on experimental data or, where data are scarce, alternative estimation methods such as multiple linear regression or a simple substitution by analogy approach. Additional bulk water activity and electrodynamic

balance measurements were carried out to augment the database for the AIOMFAC parameter fit. While not optimal, we show that the use of alternative parameter estimation methods enables physically sound predictions and offers the benefit of a more broadly applicable model. Our implementation of the aqueous carbonate/bicarbonate/$CO_{2(aq)}$ system accounts for the associated temperature-dependent dissociation equilibria explicitly and enables closed- or open-system computations with respect to carbon dioxide equilibration with the gas phase. We discuss different numerical approaches for solving the coupled equilibrium

conditions and highlight critical considerations when extremely acidic or basic mixtures are encountered.

The fitted AIOMFAC model performance for inorganic aqueous systems is considered excellent over the whole range of mixture compositions where reference data are available. Moreover, the model provides physically meaningful predictions of water activity under highly concentrated conditions. For organic–inorganic mixtures involving new species, the model–measurement agreement is found to be good in most cases, especially at equilibrium relative humidities above $\sim 70\%$; reasons for devi-

ations are discussed. Several applications of the extended model are shown and discussed, including: the effects of ignoring the auto-dissociation of water in carbonate systems, the effects of mixing bisulfate and bicarbonate compounds in closed- or open-system scenarios on pH and solution speciation, and the prediction of critical cloud condensation nucleus activation of NaI or $Na_2CO_3$ particles mixed with suberic acid.



# 1 Introduction

Atmospheric aerosols constitute a wide range of organic compounds, inorganic salts or acids, and water (e.g., Murphy and Thomson, 1997; Lee et al., 2002; Zhang et al., 2007). The semivolatile organics or electrolytes cause gas–particle partitioning which affects the physical state of the condensed phase. The presence of dissolved and/or solid phases of inorganic salts/acids (i.e., electrolytes) and their interplay with organic compounds may lead to a polar aqueous phase and a less polar organic phase (Marcolli and Krieger, 2006). For industrial applications, inorganic salts are often chosen as purifying agents between organic

compounds and water. In tropospheric aerosol particles, non-ideal interaction between electrolytes and organics may result in complex aerosol morphologies such as liquid–liquid phase separation shown by various experiments (Smith et al., 2011; Bertram et al., 2011; Song et al., 2012; Veghte et al., 2013; Altaf and Freedman, 2017; Gorkowski et al., 2017; Huang et al., 2021) and modeling results (Zuend et al., 2010; Renbaum-Wolff et al., 2016; Ovadnevaite et al., 2017; Rastak et al., 2017; Pye et al., 2018). To understand the non-ideal mixing of a wide range of inorganic and organic aerosol constituents, reliable

thermodynamic models have to be developed.

Historically, a Pitzer-based approach has been shown to describe the thermodynamics in aqueous electrolyte solutions very well up to high ionic strength (i.e., 10 molal) (Pitzer, 1991; Clegg et al., 1998a, b; Zuend et al., 2008). Substance-specific UNIQUAC (UNIversal QUAsi Chemical) (Abrams and Prausnitz, 1975) or the group–contribution version UNIFAC (UNIquac Functional group Activity Coefficients) (Fredenslund et al., 1975) is widely used in characterizing non-ideal mixing in aque-

ous organic systems. LIQUAC (Li et al., 1994) and its group–contribution version LIFAC (Yan et al., 1999) use a combination of a Pitzer ion-interaction model and UNIQUAC/UNIFAC to describe the thermodynamics of mixing in systems containing aqueous electrolytes and organic compounds. Since UNIQUAC/UNIFAC and LIFAC were originally developed for chemical engineering purposes, the selection of organic compounds and the temperature range of interest differ from those of relevance for atmospheric aerosols. Based on LIFAC, a thermodynamic model AIOMFAC (Aerosol Inorganic-Organic Mixture Func-

tional Groups Activity Coefficients) has been developed to describe non-ideal mixing in atmospheric aerosols (Zuend et al., 2008, 2011). In this study, an extension of AIOMFAC is presented with interactions of newly introduced cations ↔ anions and ions ↔ organic main groups. The extension focuses on the inclusion of iodine, carbonate electrolytes, and the associated equilibria in water and organic compound mixtures due to these species' relevance in specific environments such as the marine boundary layer or specific atmospheric condensed phases such as cloud water.

The abundance of iodine species in atmospheric aerosols is generally very small in the range from $< 0.1$ to approximately $20\ \mathrm{ng\ m^{-3}}$ (Saiz-Lopez et al., 2012). It is well established that the oceans are the dominant aerosol iodine source based on direct aerosol iodine concentration measurements in different coastal and marine locations (Saiz-Lopez et al., 2012). High elemental iodine to chlorine ($\mathrm{I/Cl}$) or iodine to sodium ($\mathrm{I/Na}$) ratios in aerosol particles in the marine atmospheric boundary layer compared to those ratios in seawater are strong evidence of oceans as the main source of iodine as well (Duce et al.,

1967; Baker et al., 2000). The observed enrichment of iodine in marine particulate matter is likely the result of the material exchange between volatile iodine species such as $\mathrm{I_2}$ in the gas phase and marine aerosols. Also, the bubble bursting process of sea spray aerosol formation can contribute to iodine species in the aerosol phase (Seto and Duce, 1972). Both inorganic





and organic iodine (e.g., $I^-$, $CH_3I$) contribute to total iodine in the aerosol phase; although, which form dominates may vary and is at present unclear. Dissolved iodide ($I^-$) and iodate ($IO_3^-$) are considered the most important inorganic forms in the

aqueous phase (Saiz-Lopez et al., 2012). From measurements across various sites, the ratio between $I^-$ and $IO_3^-$ is highly variable in different marine rainwater and aerosol samples (Saiz-Lopez et al., 2012). To date, no clear chemical pathway has been proposed to explain this variability. Although, the low ratio between $IO_3^-$ and $I^-$, especially in fine aerosols, may be partially attributed to inorganic reactions in an acidic medium or the production of $I^-$ through HOI reaction with dissolved organic matter (Pechtl et al., 2007). Other studies have shown that insoluble iodine also constitutes a significant fraction of

marine aerosols (Tsukada et al., 1987; Baker et al., 2000; Xu et al., 2010; Gilfedder et al., 2010). It is likely that part of the insoluble portion is in the form of organic species (Baker et al., 2000) or as material adsorbed to mineral or black carbon surfaces (Gilfedder et al., 2010). Moreover, there is increased interest in the iodine chemistry in the Arctic as iodine species are involved in ozone depletion and new particle formation (NPF) events (Allan et al., 2015; Raso et al., 2017; Dall´Osto et al., 2018; Baccarini et al., 2020). Different iodine species like $IO_3^-$, IO, and $I_2$ have been detected in the Arctic marine boundary

layer. Among them, gaseous iodic acid ($HIO_3$) has been identified as the primary driver for NPF events in the central Arctic Ocean during summer and fall seasons (Baccarini et al., 2020). Such events can potentially influence the number concentration and hygroscopicities of cloud condensation nuclei (CCN) and the cloud microphysical properties in the region. Since the CCN number concentration is typically limited in the Arctic, the cloud radiative effect is extremely sensitive to any perturbation in NPF events. Thus, a better understanding of the iodine chemistry would better constrain the Arctic cloud radiative forcing. In

addition, Koenig et al. (2020) have managed to quantitatively detect the presence of iodine in the stratosphere. Previously, Kim et al. (2016) investigated the thermodynamic properties in a number of alkali halide aqueous solutions including some iodide salts using a modified UNIFAC method. Other groups like Al-Sahhaf and Jabbar (1993) and Iliuta and Thyrion (1995) only studied the specific UNIQUAC interaction parameters in organic solvent + iodide salts of their interest. Through the addition of $I^-$ in AIOMFAC, we provide a first step to describing and exploring the thermodynamics in different mixtures of water +

organics + iodide electrolytes in marine aerosols. Currently, no attempts have been made on the inclusion of $IO_3^-$ ion in any thermodynamic models due to the limited availability of experimental data in aqueous iodate salts. For the same reason, only a selection of $IO_3^-$ species have been included in AIOMFAC based on experimental data at the moment as further discussed in Sect. 3.1.

Dust storms have various environmental impacts and are important sources of tropospheric aerosols (e.g., Sviridenkov et al.,

1993; Gomes and Gillette, 1993). The arid and semi-arid regions of Asia and North America, plus the Sahara desert are the three major sources of mineral dust in the northern hemisphere (Gomes and Gillette, 1993). Field measurements have reported that rich alkaline elements mainly in the form of carbonate species like $CaCO_3$, $MgCO_3$, and $K_2CO_3$ are present in collected aerosol samples (Gillette et al., 1992; Andronova et al., 1993; Gomes and Gillette, 1993). Interestingly, the alkalinity of Sahara dust may also contribute to the relatively high concentration of $IO_3^-$ in mineral dust aerosols (Baker, 2004, 2005; Allan et al.,

2009). The dust originating from the Sahara desert can be lofted and transported over large distances, with impacts on clouds and precipitation far away from the source, such as on precipitation events in Europe (Loÿe-Pilot et al., 1986). On the local scale, construction sites and vehicular movement can be major contributors to carbonate-containing airborne dust particles





(Clarke and Karani, 1992). Due to the various chemical equilibria involving carbonate species in solid or aqueous aerosol phases, the capacity of carbonate salts to neutralize acidity affects atmospheric chemical reactions and, via deposition, surface

ocean alkalinity. For example, the heterogeneous conversion process of sulfur and nitrogen oxides to particulate sulfate and nitrate is facilitated by the presence of carbonate (Dentener et al., 1996). Additionally, dust aerosols affect the global radiation budget by absorbing or scattering solar and terrestrial radiation (Tegen et al., 1996).

In previous work, Harvie et al. (1984) and Wexler and Clegg (2002) developed a section of the thermodynamic model E-AIM covering aqueous carbonate species ($HCO_3^-$, $CO_3^{2-}$, $CO_{2(aq)}$) and a set of other important ions (e.g., $Na^+$, $K^+$, $HSO_4^-$,

$SO_4^{2+}$), based on the Pitzer ion-interaction approach. Another gas–aerosol equilibrium model, SCAPE 2 (Meng et al., 1995), has been developed using the Kusik-Meissner method (Kusik and Meissner, 1978), the Pitzer ion-interaction approach, and the Zdanovskii–Stokes–Robinson (ZSR) method (Zdanovskii, 1948; Stokes and Robinson, 1966). In addition to carbonates, crustal species like $K^+$, $Mg^{2+}$, and $Ca^{2+}$ have been also included in SCAPE 2. To date, only a few groups (e.g., Xie et al., 2018; Fu et al., 2019) have investigated the liquid–liquid equilibrium thermodynamics of systems composed of a carbonate salt +

water + organic; also, the organic compound selection in previous work was limited to specific systems of interest in chemical purification processes. By adding carbonates to AIOMFAC, the model allows the description of multi-ion aqueous organic–inorganic systems and acidity effects of $HCO_3^-$, $CO_3^{2-}$, and dissolved carbon dioxide ($CO_{2(aq)}$). A further motivation for including the outlined new inorganic species in the AIOMFAC model is to enable model–measurement comparisons, including recent cloud condensation nuclei (CCN) activation experiments by Davies et al. (2019). Such model applications are discussed

in Sect. 5.2.

## 2   Methods

### 2.1   AIOMFAC model framework

The degree of non-ideality in a multicomponent system is represented by the activity coefficients of all components within the system's (liquid) phases. To describe the concentration of each constituent in a system, mole fraction, $x_j = n_j/(\sum_s n_s + $

$\sum_i n_i)$, is often used. Here, $n_j$ is the molar amount of species $j$, the summation index $s$ covers all non-ion ("solvent") components, while $i$ covers inorganic ions; therefore, this definition of mole fraction is with respect to partially or completely dissociated ions (in other cases the definition may differ). For ions, concentration is often expressed in terms of molality (moles of ion $i$ per unit mass of solvent or solvent mixture), $m_i = n_i/(\sum_s n_s M_s)$ with $M_s$ ($mol\,kg^{-1}$) being the molar mass of compound $s$. Within AIOMFAC, molality is defined on the basis of the solvent mixture (water and organic compounds). In

AIOMFAC, water and organic compound activities ($a_s^{(x)}$) are defined on a mole fraction basis (as denoted here by superscript $(x)$), a typical choice for non-ionic species. In contrast, ion activity ($a_i^{(m)}$) and activity coefficient ($\gamma_i^{(m)}$) are defined on a molality basis (superscript $(m)$), with pure water as the reference solvent. For simplicity of notation, these superscripts will be omitted hereafter. In experiments, it is impossible to determine the activity coefficient of a single ion. Hence, the mean molal



activity coefficient ($\gamma_\pm$) of an electrolyte is used instead:

$$\gamma_\pm = \left[ \gamma_+^{v^+} \cdot \gamma_-^{v^-} \right]^{1/(v^+ + v^-)}, \tag{1}$$

where $v^+$ and $v^-$ are the stoichiometric numbers of the charge-balanced electrolyte's cations (+) and anions (−), respectively. In the case of aqueous $CO_{2(aq)}$, unlike other non-electrolytes, its activity coefficient is also defined on molality basis within our AIOMFAC extension, as described in Sect. 3.2.1.

The deviation from mixing ideality within a thermodynamic system is characterized by the Gibbs excess energy ($G^{ex}$), from which expressions for activity coefficients can be derived. In AIOMFAC, $G^{ex}$ is composed of long-range (LR), middle-range (MR), and short-range (SR) contributions effectively accounting for the various Coulombic and van der Waals interactions among solution species. The hydrated relative group volume $R_t^H$ and the relative surface area $Q_t^H$ parameters used in the short-range part describing the physical properties of the newly introduced ions are listed in Table 1. The theoretical framework of the AIOMFAC model has been discussed in detail in prior work (Zuend et al., 2008, 2011). In the following, we focus on selected model expressions of relevance for the implementation and parameterization of the new species, which mainly concerns the middle-range contributions and associated interactions among water, organic main groups, and inorganic ions. The middle-range contribution to $G^{ex}$ in a mixture of $n$ moles of main groups $k$ and of ions $i$ is (Zuend et al., 2008)

$$
\begin{aligned}
\frac{G_{MR}^{ex}}{RT} =\ & \frac{1}{\sum_k n_k M_k} \sum_k \sum_i B_{k,i}(I) n_k n_i + \frac{1}{\sum_k n_k M_k} \sum_c \sum_a B_{c,a}(I) n_c n_a \\
& + \frac{1}{\sum_k n_k M_k} \sum_c \sum_a C_{c,a}(I) n_c n_a \sum_i \frac{n_i |z_i|}{\sum_k n_k M_k} + \frac{1}{\sum_k n_k M_k} \sum_c \sum_{c' \geq c} R_{c,c'} n_c n_{c'} \\
& + \frac{1}{(\sum_k n_k M_k)^2} \sum_c \sum_{c' \geq c} \sum_a Q_{c,c',a} n_c n_{c'} n_a.
\end{aligned} \tag{2}
$$

Here, the subscripts $c$ and $a$ stand for cations and anions, respectively, and $I$ ($\mathrm{mol\,kg^{-1}}$) is the molal ionic strength of the phase. $B_{k,i}(I)$ are ionic-strength-dependent functions describing interactions among (solvent) main groups and ions. $B_{c,a}(I)$ and $C_{c,a}(I)$ are parameters accounting for interactions among specific cation–anion pairs in aqueous solution. $R_{c,c'}$ and $Q_{c,c',a}$ were introduced to account for binary cation–cation or ternary cation–cation–anion interactions in the presence of $NH_4^+$, $H^+$ or $NH_4^+$, $H^+$, $SO_4^{2-}$ specifically. Without those additional interaction terms considered, the AIOMFAC model would be unable to match measured $a_w$ and dissociation degree data at low water contents; see discussion in Zuend et al. (2008). While in case of other systems, explicit consideration of such interactions, while present, is considered unnecessary. Since the reference solvent (for ions) in AIOMFAC is water, the $B_{k,i}$ contributions are defined to be zero for all water–ion interactions. The first three ionic-strength-dependent functions in Eq. (2) are parameterized by a functional form typical for Pitzer-like models (Zuend



**Table 1.** Physical properties of the new ions introduced in the study.

| Ion | $r_c$ (pm) [a] | $N^{\mathrm{ADH}}$ [b] | $R_t$ | $Q_t$ | $R_t^H$ [c] | $Q_t^H$ [c] |
|---|---|---|---|---|---|---|
| $I^-$ | 220 | 0.00 | 1.77 | 1.47 | 1.77 | 1.47 |
| $IO_3^-$ | 209 | 0.00 | 1.52 | 1.32 | 1.52 | 1.32 |
| $HCO_3^-$ | 185 | 2.00 | 1.05 | 1.04 | 2.89 | 3.84 |
| $CO_3^{2-}$ | 178 | 4.00 | 0.94 | 0.96 | 4.62 | 6.56 |
| $OH^-$ | 135 | 2.80 | 0.41 | 0.56 | 2.99 | 4.47 |

[a] The unhydrated radius of $OH^-$ is taken from Kiriukhin and Collins (2002), while the other radii are from Marcus (1994).
[b] The apparent dynamic hydration number ($N^{\mathrm{ADH}}$) of $OH^-$ is taken from Kiriukhin and Collins (2002) with $HCO_3^-$ and $CO_3^{2-}$ from Marcus (1994).
[c] Detailed description of the calculation process was discussed in Zuend et al. (2008).

et al., 2008):

$$B_{k,i}(I) = b_{k,i}^{(1)} + b_{k,i}^{(2)} e^{-b_{k,i}^{(3)}\sqrt{I}}, \tag{3}$$

$$B_{c,a}(I) = b_{c,a}^{(1)} + b_{c,a}^{(2)} e^{-b_{c,a}^{(3)}\sqrt{I}}, \tag{4}$$

$$C_{c,a}(I) = c_{c,a}^{(1)} e^{-c_{c,a}^{(2)}\sqrt{I}}, \tag{5}$$

with $b$ and $c$ being adjustable AIOMFAC parameters. The parameters $b_{k,i}^{(3)}$ are always kept as 1.2 kg$^{1/2}$mol$^{-1/2}$, while $b_{c,a}^{(3)}$ are set to 0.8 kg$^{1/2}$mol$^{-1/2}$ by default and only changed in a few cases to achieve better agreement between the model and experimental data. All binary MR interaction functions are symmetric, i.e., $C_{c,a}(I) = C_{a,c}(I)$. Introduction of new ions or organic functional groups into the AIOMFAC model requires the determination of the relevant parameters from Eqs. (3)–(5), usually via a AIOMFAC model fit to adequate experimental data.

## 2.2 Objective function

To obtain the relevant parameters, an objective function is derived to allow the direct comparison between experimental data and model calculations. The objective function, subject to minimization during the simultaneous fitting process concerning parameters for either a cation–anion pair or ion–organic main group is

$$F_{\mathrm{obj}} = \sum_d \sum_u w_{d,u} \ln\left[\frac{|Q_{d,u}^{\mathrm{calc}}| + Q_{d,u}^{\mathrm{tol}}}{|Q_{d,u}^{\mathrm{ref}}| + Q_{d,u}^{\mathrm{tol}}}\right]^2. \tag{6}$$

Here, $u$ is a specific data point in the dataset $d$ of assigned weight $w_{d,u}$. $Q_{d,u}^{\mathrm{calc}}$ and $Q_{d,u}^{\mathrm{ref}}$ are model predicted and experimentally determined thermodynamic quantities accordingly. $Q_{d,u}^{\mathrm{tol}}$ is the tolerance quantity, which represents the estimated model sensitivity for the targeted quantity, as described by Zuend et al. (2011). During the fitting process, an additional constraint is applied on the resulting model curve of water activity. Consequently, the water activity is forced to decrease monotonically



with increasing ionic strength for reasonable extrapolation at supersaturation conditions. The treatment of different data types and associated constraints is discussed in Zuend et al. (2011) and those procedures were also adopted for this study.

## 3    New systems: properties, chemical equilibria, and data types

As described previously by Zuend et al. (2008, 2011), all the inorganic salts and acids are assumed to dissociate completely
into ions in the liquid mixture, except for the diprotic sulfuric and carbonic acids and related bisulfate/bicarbonate salts. The relevant equilibria and dissociation steps concerning sulfuric and carbonic acids are dealt with explicitly. We note that in reality some degree of ion association (such as temporary or permanent ion pairs in aqueous solutions) is likely present, especially in solutions of higher electrolyte concentrations. In the AIOMFAC approach, such ion association effects are implicitly factored (fitted) into the activity coefficients of the solvents and solutes.

### 3.1    Inorganic iodine system

#### 3.1.1    Data sources

For the mixture of inorganic iodide ($I^-$) salts or acids with water, a large number of data sources are available covering the water activity and mean molal activity coefficient of the ions from low to high ionic strength. In contrast, only very few datasets are available for iodate ($IO_3^-$) electrolytes and most of them cover only the very dilute aqueous concentration range
(i.e., 0.3 molal). Some ternary solubility data for relatively low $IO_3^-$ molalities in solutions dominated by other salts (of total ionic strength 3 molal) are also present. To validate and complement the existing iodate data sources, we performed our own bulk water activity measurements for binary aqueous $NaIO_3$, $KIO_3$, and $HIO_3$ solutions, as well as aqueous solutions of a selection of ternary salt mixtures, which are tabulated in the Supplemental Information (SI), Sect. S4. Table 2 lists the aqueous iodide or iodate electrolytes, data types, and references that were used in fitting associated AIOMFAC parameters. Most of the
experiments were conducted at room temperature, i.e., around 298.15±5 K, except for measurements at 313.15 K by Patil et al. (1990). Since activity coefficients (and activities) are weak functions of temperature, the exact temperature of the experiment is not critical for determining AIOMFAC parameters.

In terms of organic compounds and their interaction with iodides, several groups (e.g., Nasehzadeh et al., 2004; Mato and Cocero, 1988) have investigated the mixing thermodynamics of inorganic iodide salts and alcohols. After the initial data
comparison, we decided to dismiss the data reported by Yamamoto et al. (1997), since their VLE composition data contain some ambiguity and show significant deviation from other measurements. Measurements covering carboxylic acids are scarce; we found only data on the hygroscopicity of internally mixed $NaI$ + succinic acid particles by Miñambres et al. (2011). By comparing some of their binary salt + water experimental mass growth factor data to our AIOMFAC predictions (considered to be accurate and well-determined by other data), it is evident that there are unexplained deviations in their reported mass
growth factors. Therefore, we performed several sets of bulk or electrodynamic balance (EDB) water activity measurements for systems composed of sodium iodide, water and different carboxylic acids, with the associated measurement data tabulated





**Table 2.** Data type, number of points ($N_d$), and references for data used to fit the MR interaction parameters of cation–anion pairs in aqueous iodide or iodate systems.

| Electrolytes | Data type [a] | $N_d$ | Reference [b] |
|---|---|---|---|
| | Binary mixtures (1 salt/acid + water) | | |
| NaI | $\gamma_\pm$ | 4 | Gregoriou et al. (1979) |
| | $\gamma_\pm$ | 35 | Hamer and Wu (1972) |
| | $\gamma_\pm$ | 4 | Harned (1929) |
| | $\gamma_\pm$ | 21 | Pan (1981) |
| | $\gamma_\pm$ | 11 | Robinson (1935) |
| | $\gamma_\pm$ | 18 | Robinson and Stokes (1949) |
| | $\gamma_\pm$ | 68 | Zaytsev and Aseyev (1992) |
| | $a_w$ | 35 | Hamer and Wu (1972) |
| | $a_w$ | 21 | Pan (1981) |
| | $a_w$ | 18 | Robinson and Stokes (1949) |
| KI | $\gamma_\pm$ | 11 | Gelbach (1933) |
| | $\gamma_\pm$ | 26 | Hamer and Wu (1972) |
| | $\gamma_\pm$ | 7 | Harned (1929) |
| | $\gamma_\pm$ | 21 | Pan (1981) |
| | $\gamma_\pm$ | 29 | Partanen (2010) |
| | $\gamma_\pm$ | 18 | Pearce and Nelson (1932) |
| | $\gamma_\pm$ | 13 | Robinson (1935) |
| | $\gamma_\pm$ | 20 | Robinson and Stokes (1949) |
| | $\gamma_\pm$ | 13 | Robinson et al. (1940) |
| | $\gamma_\pm$ | 42 | Zaytsev and Aseyev (1992) |
| | $a_w$ | 26 | Hamer and Wu (1972) |
| | $a_w$ | 21 | Pan (1981) |
| | $a_w$ | 29 | Partanen (2010) |
| | $a_w$ | 18 | Pearce and Nelson (1932) |
| | $a_w$ | 20 | Robinson and Stokes (1949) |
| HI | $\gamma_\pm$ | 33 | Hamer and Wu (1972) |
| | $\gamma_\pm$ | 15 | Harned (1929) |
| | $\gamma_\pm$ | 13 | Hetzer et al. (1964) |
| | $\gamma_\pm$ | 5 | Kielland (1937) |
| | $\gamma_\pm$ | 21 | Pan (1981) |
| | $\gamma_\pm$ | 17 | Robinson and Stokes (1949) |
| | $a_w$ | 33 | Hamer and Wu (1972) |
| | $a_w$ | 15 | Harned (1929) |
| | $a_w$ | 21 | Pan (1981) |
| | $a_w$ | 17 | Robinson and Stokes (1949) |
| NH$_4$I | $\gamma_\pm$ | 29 | Bonner (1976) |
| | $\gamma_\pm$ | 17 | Robinson and Stokes (1949) |
| | $\gamma_\pm$ | 29 | Zaytsev and Aseyev (1992) |
| | $\gamma_\pm$ | 10 | Revie and Uhlig (2008) |
| | $a_w$ | 29 | Bonner (1976) |



| Electrolytes | Data type [a] | $N_d$ | Reference [b] |
|---|---|---|---|
| \multicolumn{4}{c}{Binary mixtures (1 salt/acid + water)} | | | |
| LiI | $\gamma_\pm$ | 23 | Hamer and Wu (1972) |
| | $\gamma_\pm$ | 21 | Pan (1981) |
| | $\gamma_\pm$ | 10 | Robinson and Sinclair (1934) |
| | $\gamma_\pm$ | 17 | Robinson and Stokes (1949) |
| | $a_w$ | 23 | Hamer and Wu (1972) |
| | $a_w$ | 21 | Pan (1981) |
| | $a_w$ | 6 | Patil et al. (1990) |
| | $a_w$ | 17 | Robinson and Stokes (1949) |
| $MgI_2$ | $\gamma_\pm$ | 45 | Goldberg and Nuttall (1978) |
| | $\gamma_\pm$ | 12 | Mullin (2001) |
| | $\gamma_\pm$ | 9 | Robinson and Harned (1941) |
| | $\gamma_\pm$ | 13 | Robinson et al. (1940) |
| | $\gamma_\pm$ | 21 | Stokes (1948) |
| | $\gamma_\pm$ | 43 | Zaytsev and Aseyev (1992) |
| | $a_w$ | 45 | Goldberg and Nuttall (1978) |
| | $a_w$ | 13 | Robinson et al. (1940) |
| $CaI_2$ | $\gamma_\pm$ | 38 | Goldberg and Nuttall (1978) |
| | $\gamma_\pm$ | 8 | Mullin (2001) |
| | $\gamma_\pm$ | 15 | Stokes (1948) |
| | $\gamma_\pm$ | 38 | Zaytsev and Aseyev (1992) |
| | $a_w$ | 38 | Goldberg and Nuttall (1978) |
| | $a_w$ | 15 | Stokes (1948) |
| $NaIO_3$ | $\gamma_\pm$ | 6 | Durig et al. (1965) |
| | $a_w$ | 6 | Durig et al. (1965) |
| | $a_w$ | 4 | this study (bulk) |
| $KIO_3$ | $\gamma_\pm$ | 3 | Bonner and Prichard (1979) |
| | $a_w$ | 3 | Bonner and Prichard (1979) |
| | $a_w$ | 1 | Apelblat and Korin (1998) [c] |
| | $a_w$ | 4 | this study (bulk) |
| $LiIO_3$ | $\gamma_\pm$ | 18 | Bonner and Prichard (1979) |
| | $a_w$ | 18 | Bonner and Prichard (1979) |
| $HIO_3$ | $\gamma_\pm$ | 19 | Abel et al. (1934) |
| | $\gamma_\pm$ | 14 | Pethybridge and Prue (1967) |
| | $\gamma_\pm$ | 12 | Durig et al. (1965) |
| | $\gamma_\pm$ | 14 | Goldman et al. (1974) |
| | $a_w$ | 12 | Durig et al. (1965) |
| | $a_w$ | 14 | Goldman et al. (1974) |
| | $a_w$ | 17 | Kumar et al. (2010) [c] |
| | $a_w$ | 94 | Murray et al. (2012) [d] |
| | $a_w$ | 14 | this study (bulk) |





| Electrolytes | Data type [a] | $N_d$ | Reference [b] |
|---|---|---|---|
| *Multicomponent mixtures* [e] | | | |
| $NaBr + NaIO_3 \,[17.88:1]$ | $a_w$ | 4 | this study (bulk) |
| $KBr + KIO_3 \,[10.75:1]$ | $a_w$ | 4 | this study (bulk) |
| $NaIO_3 + NaBr$ | SLE | 5 | Ricci (1934) |
| $KIO_3 + KBr$ | SLE | 6 | Ricci (1934) |
| $KIO_3 + KI$ | SLE | 6 | Ricci (1937) |
| $KIO_3 + KCl$ | SLE | 6 | Hill and Ricci (1931) |
| $Mg(IO_3)_2 + Mg(NO_3)_2$ | SLE | 8 | Hill and Ricci (1931) |

[a] Water activity ($a_w$) data were calculated from osmotic coefficients if not provided in the literature.
[b] All the experiments were done at 298.15±5 K except for those by Patil et al. (1990) at 313.15 K.
[c] The weighting is set to zero for the fitting of model parameters due to large disagreement with our own measurement.
[d] The weighting is reduced for the fitting of model parameters due to its composition calibration to that from Kumar et al. (2010).
[e] Stated in the brackets are the corresponding molar ratios of the salt mixtures.

in Sect. S4 of the SI. In our EDB measurement on water + citric acid + NaI particles, when the RH was decreased to below 40 %, complete equilibration of the droplet with the gas phase inside the EDB trap may have been substantially impeded due to the high viscosity of citric acid at low water contents. Hence, only data for RH > 40 % were considered for the AIOMFAC

fit.

In the past, measurements involving methanol were generally excluded from the AIOMFAC parameter optimization procedure. Being the shortest chain alcohol, methanol tends to behave differently than a simple extrapolation from longer-chain alcohols would suggest (e.g., well-known for saturation vapor pressures (Donahue et al., 2011)). This may affect the determined parameter describing interactions between the hydroxyl group OH and $I^-$, meant to be applicable to a wide range

of alcohols/polyols and other compounds. However, due to the scarcity of experimental data, available data for components containing organic main groups like aromatic hydrocarbon (ACH) are mainly present in water-free ternary systems involving methanol, e.g., benzene + methanol + KI. To have a wider coverage of data supporting the fit of $I^- \leftrightarrow$ organic main group parameters, we included some datasets with methanol. Table 3 lists all the datasets used in the simultaneous fit of new AIOMFAC parameters.

**3.2   Carbonic acid system**

In the presence of carbonic acid, there are four relevant equilibria to consider,

$$CO_{2(aq)} + H_2O \;\rightleftharpoons\; HCO_3^- + H^+, \tag{R1}$$

$$HCO_3^- \;\rightleftharpoons\; H^+ + CO_3^{2-}, \tag{R2}$$

$$H_2O \;\rightleftharpoons\; H^+ + OH^-, \tag{R3}$$

$$CO_{2(g)} \;\rightleftharpoons\; CO_{2(aq)}. \tag{R4}$$





**Table 3.** Data summary of iodide salts mixed with organic compounds and thereby various organic main groups in aqueous solutions (unless specified otherwise).

| Organic compounds | Org. main groups | Salts | $T$ (K) | Data type | $N_d$ | $w_d^{\text{init}}$ | Reference |
|---|---|---|---|---|---|---|---|
| $-$ *water + alcohol/sugar + salt systems* $-$ | | | | | | | |
| ethanol (water-free) | $CH_n$, $CH_n^{[OH]}$, OH | LiI | 298 | VLE(org) | 13 | 0.50 | Safarov (2005) |
| ethanol (water-free) | $CH_n$, $CH_n^{[OH]}$, OH | LiI | 298 | VLE(org) | 20 | 1.00 | Nasehzadeh et al. (2004) |
| ethanol (water-free) | $CH_n$, $CH_n^{[OH]}$, OH | NaI | 298 | VLE(org) | 65 | 1.00 | Nasehzadeh et al. (2004) |
| ethanol (water-free) | $CH_n$, $CH_n^{[OH]}$, OH | NaI | 298 | VLE(org) | 6 | 1.00 | Mato and Cocero (1988) |
| ethanol (water-free) | $CH_n$, $CH_n^{[OH]}$, OH | NaI | 298 | VLE(org) | 16 | 1.00 | Barthel and Lauermann (1986) |
| ethanol | $CH_n$, $CH_n^{[OH]}$, OH | NaI | 298 | SLE | 12 | 0.05 | Pawar et al. (2012) |
| 1-propanol | $CH_n$, $CH_n^{[OH]}$, OH | NaI | 298 | SLE | 9 | 0.10 | Kacperska (1994) |
| 2-propanol | $CH_n$, $CH_n^{[OH]}$, OH | NaI | 298 | SLE | 13 | 0.05 | Pawar et al. (2012) |
| sorbitol | $CH_n^{[OH]}$, OH | NaI | 298 | $a_w$(bulk) | 7 | 5.00 | this study |
| sorbitol | $CH_n^{[OH]}$, OH | NaI | 298 | $a_w$(EBD) | 10 | 1.00 | this study |
| D-mannopyranose | $CH_n^{[OH]}$, OH,$CH_n$O | NaI | 298 | $\gamma_\pm$ | 32 | 2.00 | Yang et al. (2004) |
| D-ribofuranose | $CH_n^{[OH]}$, OH,$CH_n$O | NaI | 298 | $\gamma_\pm$ | 32 | 2.00 | Yang et al. (2004) |
| maltose | $CH_n^{[OH]}$, OH,$CH_n$O | NaI | 298 | $\gamma_\pm$ | 35 | 2.00 | Zhuo et al. (2008) |
| ethanol | $CH_n$, $CH_n^{[OH]}$, OH | KI | 298 | VLE | 15 | 1.00 | Burns and Furter (1979) |
| ethanol | $CH_n$, $CH_n^{[OH]}$, OH | KI | 298 | VLE | 21 | 1.00 | Sun (1996) |
| ethanol | $CH_n$, $CH_n^{[OH]}$, OH | KI | 298 | VLE | 28 | 1.00 | Chen and Zhang (2003) |
| 1-propanol | $CH_n$, $CH_n^{[OH]}$, OH | KI | 298 | VLE | 8 | 1.00 | Chen and Zhang (2003) |
| 1-propanol | $CH_n$, $CH_n^{[OH]}$, OH | KI | 298 | VLE | 11 | 1.00 | Yin et al. (2018) |
| 1-propanol | $CH_n$, $CH_n^{[OH]}$, OH | KI | 298 | SLE | 11 | 0.10 | Kacperska (1994) |
| 1-propanol | $CH_n$, $CH_n^{[OH]}$, OH | KI | 298 | SLE | 11 | 0.05 | Pawar et al. (2010) |
| 2-propanol | $CH_n$, $CH_n^{[OH]}$, OH | KI | 298 | VLE | 10 | 1.00 | Chen and Zhang (2003) |
| 2-propanol | $CH_n$, $CH_n^{[OH]}$, OH | KI | 298 | SLE | 8 | 1.00 | Il'In and Cherkasov (2016) |
| 1-butanol | $CH_n$, $CH_n^{[OH]}$, OH | KI | 298 | LLE | 15 | 1.00 | Al-Sahhaf and Kapetanovic (1997) |
| $-$ *water + carboxylic acid + salt systems* $-$ | | | | | | | |
| malonic acid | $CH_n$, COOH | NaI | 298 | $a_w$(bulk) | 25 | 5.00 | this study |
| glutaric acid | $CH_n$, COOH | NaI | 298 | $a_w$(bulk) | 5 | 5.00 | this study |
| glutaric acid | $CH_n$, COOH | NaI | 298 | $a_w$(EDB) | 59 | 2.00 | this study |
| citric acid | $CH_n$, COOH | NaI | 298 | $a_w$(bulk) | 6 | 5.00 | this study |
| citric acid | $CH_n$, COOH | NaI | 298 | $a_w$(EDB) | 38 | 2.00 | this study |
| $-$ *water + ketone + salt systems* $-$ | | | | | | | |
| acetone, methanol (water-free) | $CH_n$, $CH_n$CO | NaI | 298 | VLE | 108 | 1.00 | Iliuta and Thyrion (1995) |
| acetone | $CH_n$, $CH_n$CO | NaI | 298 | SLE | 4 | 1.00 | Macy and Thomas (1926) |
| acetone | $CH_n$, $CH_n$CO | KI | 298 | VLE | 34 | 0.10 | Al-Sahhaf and Jabbar (1993) |
| acetone, methanol (water-free) | $CH_n$, $CH_n$CO | KI | 298 | VLE | 44 | 1.00 | Iliuta and Thyrion (1995) |
| 2-butanone | $CH_n$, $CH_n$CO | KI | 298 | LLE | 10 | 0.02 | Al-Sahhaf et al. (1999) |
| $-$ *alcohol + ether + salt systems* $-$ | | | | | | | |
| 1,4-dioxane | $CH_n$, CHnO | KI | 298 | VLE | 43 | 0.50 | Liu et al. (1998) |
| $-$ *water + ester + salt systems* $-$ | | | | | | | |
| ethyl acetate | $CH_n$, CCOO | NaI | 298 | SLE | 5 | 1.00 | Altshuller and Everson (1953) |
| ethyl acetate | $CH_n$, CCOO | NaI | 298 | SLE | 6 | 1.00 | Glasstone et al. (1926) |
| ethyl acetate | $CH_n$, CCOO | KI | 298 | LLE | 10 | 0.02 | Al-Sahhaf et al. (1999) |
| ethyl acetate | $CH_n$, CCOO | KI | 298 | SLE | 5 | 1.00 | Altshuller and Everson (1953) |





| Organic compounds | Org. main groups | Salts | $T$ (K) | Data type | $N_d$ | $w_d^{\text{init}}$ | Reference |
|---|---|---|---|---|---|---|---|
| $-\ water + multifunctional\ aromatic\ compounds + salt\ systems\ -$ | | | | | | | |
| benzene, methanol (water-free) | ACH$_n$ | NaI | 298 | VLE | 69 | 1.00 | Yang et al. (2007) |
| benzene, methanol (water-free) | ACH$_n$ | NaI | 298 | VLE | 60 | 1.00 | Sun (1985) |
| benzene | ACH$_n$ | NaI | 298 | SLE | 8 | 1.00 | Janado et al. (1983) |
| benzoic acid | ACH$_n$, COOH | NaI | 308 | SLE | 8 | 1.00 | Goeller and Osol (1937) |
| benzene, methanol (water-free) | ACH$_n$ | KI | 298 | VLE | 8 | 1.00 | Sun et al. (2000) |
| benzene | ACH$_n$ | KI | 298 | SLE | 10 | 1.00 | Janado et al. (1983) |
| benzoic acid | ACH$_n$, COOH | KI | 308 | SLE | 8 | 1.00 | Goeller and Osol (1937) |
| 2-hydroxybenzoic acid | ACH$_n$, ACOH, COOH | KI | 298 | SLE | 8 | 1.00 | Sugunan and Thomas (1995) |

Unlike aqueous sulfuric acid, whose first dissociation step is essentially complete (Young et al., 1959; Seinfeld and Pandis, 1998), carbonic acid's first (R1) and second (R2) dissociation steps have to be taken into account explicitly. Since carbonic acid is a strong buffer agent at nearly neutral conditions, the effects from the auto-dissociation of water (R3) on acidity are of interest as well. Finally, the Henry's law constant characterizes the partial pressure of $CO_{2(g)}$ in equilibrium with dissolved

$CO_{2(aq)}$ (R4).

A thermodynamic equilibrium constant can be expressed as

$$K = \prod_j a_j^{v_j}, \tag{7}$$

where $a_j$ and $v_j$ are the activity and the stoichiometric coefficient of species $j$ in an associated reaction equilibrium. Therefore, the equilibrium constants (which are functions of temperature and pressure) of the reactions (R1–R4) are

$$K_1 = \frac{m_{\text{H}^+} m_{\text{HCO}_3^-} \gamma_{\text{H}^+} \gamma_{\text{HCO}_3^-}}{m^\circ\, m_{\text{CO}_{2(aq)}} \gamma_{\text{CO}_{2(aq)}} a_w}, \tag{8}$$

$$K_2 = \frac{m_{\text{H}^+} m_{\text{CO}_3^{2-}} \gamma_{\text{H}^+} \gamma_{\text{CO}_3^{2-}}}{m^\circ\, m_{\text{HCO}_3^-} \gamma_{\text{HCO}_3^-}}, \tag{9}$$

$$K_w = \frac{m_{\text{H}^+} m_{\text{OH}^-} \gamma_{\text{H}^+} \gamma_{\text{OH}^-}}{(m^\circ)^2\, a_w}, \tag{10}$$

$$K_H = \frac{m_{\text{CO}_{2(aq)}} \gamma_{\text{CO}_{2(aq)}}}{m^\circ\, p_{\text{CO}_2}/p^\circ}. \tag{11}$$

Here, $p_{\text{CO}_2}$ (atm) is the partial pressure of $CO_2$, $p^\circ$ is the reference pressure in the same units (here 1 atm) and $a_w$ is the mole-

fraction-based water activity. $m_i$ and $\gamma_i$ are the molality and corresponding activity coefficient on the molality scale of ion $i$ or $CO_{2(aq)}$. The unit molality, $m^\circ = 1\ \text{mol kg}^{-1}$, is included in the above equations to highlight that equilibrium constants are dimensionless quantities (although they are scale-dependent). Since the reference values of $K$ are temperature-dependent, a





parameterization of the following functional form at a total pressure of 1 atm is often used

$$\ln(K) = p_1 + p_2 T + p_3 T^2 + p_4 T^3 + p_5/T + p_6 \ln(T) + p_7/T^2, \tag{12}$$

with the parameters $(p_1, \ldots, p_7)$ listed in Table 4 for each chemical equilibrium based on data from Marion (2001). Similar to the treatment of the partial $HSO_4^-$ dissociation in the AIOMFAC model (Zuend et al., 2008), the degree of dissociation of the $HCO_3^-$ ion, $\alpha_{HCO_3^-}$, can be expressed by

$$\alpha_{HCO_3^-} = 1 - \frac{m_{HCO_3^-}}{m_{HCO_3^-}^{max}} = 1 - \frac{n_{HCO_3^-}}{n_{HCO_3^-}^{max}}, \tag{13}$$

where $m_{HCO_3^-}$ or $n_{HCO_3^-}$ are the molality or molar amount of the $HCO_3^-$ ion and the superscript max indicates the maximum
possible amount of this ion being formed within the system for a given overall liquid-phase composition. From the equilibrium relation (R1), under the assumption that water is always more abundant in molar amount than $CO_{2(aq)}$, the molar balances of the system can be formulated as

$$n_{CO_3^{2-}}^{max} = n_{CO_{2(aq)}} + n_{CO_3^{2-}} + n_{HCO_3^-}, \tag{14}$$

$$n_{H^+}^{max} = n_{H^+} + n_{HCO_3^-} + 2n_{CO_{2(aq)}} + n_{OH^-}^{max} - n_{OH^-}. \tag{15}$$

The $OH^-$ ion is not part of the original AIOMFAC model, and while insignificant in amount and effect in acidic aerosol solutions, it was added to the model for explicit treatment in the context of the bicarbonate system and other systems of neutral to high pH. In our implementation, $n_{OH^-}^{max}$ not only represents the maximum amount of the $OH^-$ but also the fraction $(r)$ of present water considered for potential $OH^-$, $H^+$ contributions via the auto-dissociation of $H_2O$

$$n_{OH^-}^{max} = r \cdot n_{H_2O}^{init} + n_{OH^-}^{init}. \tag{16}$$

To ensure that a complete neutralization of the cumulative positive charges of cations is possible, $n_{OH^-}^{max}$ is usually set as 3.5 times the sum of the initial amount of cations. Based on Eqs. (14) and (15), the maximum molar amount of the bicarbonate ion is

$$n_{HCO_3^-}^{max} = \min\left[n_{H^+}^{max}, \, n_{CO_3^{2-}}^{max}\right]. \tag{17}$$

The equilibrium constant expressions for $K_1, K_2$, and $K_w$ (right-hand sides of Eqs 8–10) in AIOMFAC are then solved by
iteratively adjusting the molar amounts of all species until the known temperature-dependent values for different reactions as well as the molar balances (Eqs. 14 and 15) are fulfilled simultaneously. Two different numerical solving procedures were developed with their own advantages and challenges discussed in detail in SI Sect. S3.





In a system of $H_2CO_3 + Na_2CO_3 + H_2O$, there are five ions present: $H^+$, $Na^+$, $HCO_3^-$, $CO_3^{2-}$, and $OH^-$. Therefore, the AIOMFAC computation is affected by the interaction parameters of the 6 cation–anion pairs involved. Using first a few binary

systems, each only containing one cation and one anion in substantial amounts (e.g., aqueous $Na_2CO_3$ assumed to only contain $Na^+$ and $CO_3^{2-}$ ions), the fits of the model parameters for $H^+ \leftrightarrow OH^-$, $Na^+ \leftrightarrow CO_3^{2-}$, and $Na^+ \leftrightarrow OH^-$ were carried out independently. The remaining 12 middle-range interaction parameters affecting the mentioned 5-ion system, including those for $H^+ \leftrightarrow HCO_3^-$ and $H^+ \leftrightarrow CO_3^{2-}$ interactions, had to be fitted simultaneously while fulfilling the equilibria (R1)–(R4). Aqueous mixtures of $H_2CO_3 + Na_2CO_3$ of different mixing ratios were selected to better constrain the model fit.

In the presence of both dissolved carbonic acid and sulfuric acid, the bisulfate ion needs to be accounted for when the maximum possible molar amount of $H^+$ is computed,

$$n_{H^+}^{\max} = n_{H^+} + n_{HCO_3^-} + 2n_{CO_{2(aq)}} + n_{OH^-}^{\max} - n_{OH^-} + n_{HSO_4^-}. \tag{18}$$

One additional equilibrium relationship to account for the incomplete bisulfate dissociation is solved simultaneously with reactions (R1–R4)

$$HSO_4^- \rightleftharpoons H^+ + SO_4^{2-}. \tag{R5}$$

After an equilibrium composition has been established for a given input of mixture components, the acidity (pH) of the solution can be calculated as

$$pH = -\log_{10}(a_{H^+}) = -\log_{10}\left(\frac{m_{H^+}}{m^\circ}\gamma_{H^+}\right). \tag{19}$$

Some input compositions require additional considerations for meaningful calculations. For example, most computations out-

lined above are carried out assuming that all components are mixed in a liquid phase and the potential precipitation of crystalline solids (e.g., salts) is ignored (i.e., metastable, supersaturated salt solutions allowed) unless specific solid–liquid equilibria are targeted. However, there is one exception: when $Ca^{2+}$ and $SO_4^{2-}$ ions are present in a system, complete precipitation of solid $CaSO_4$ (or associated hydrates) is typically assumed to occur. This treatment of $CaSO_4$ is due to its very low solubility in water. Also, assuming that the maximum possible amount of $CaSO_4$ forms a solid instead of solving the actual solid–liquid

equilibrium of this salt allows for a more efficient AIOMFAC calculation without any loss of accuracy under most RH conditions.

### 3.2.1 Activity coefficients of $CO_{2(aq)}$

Harvie et al. (1984) have shown that the inclusion of $CO_{2(aq)}$ is essential to describe the thermodynamic behavior of the carbonic system. Following the approach taken in previous studies (Harvie et al., 1984; Clegg et al., 1991; Meng et al., 1995),





**Table 4.** Parameterization of the temperature-dependent equilibrium constants for the aqueous dissociation equilibria (R1–R4), applicable in the stated temperature range at pressures near 1 $\mathrm{atm}$ ($\sim 10^5$ Pa).

| Constant | $\ln(K)$ (298.15 K) | $p_1$ ($\times 10^2$) | $p_2$ ($\times 10^{-1}$) | $p_3$ ($\times 10^{-3}$) | $p_4$ ($\times 10^{-6}$) | $p_5$ ($\times 10^4$) | $p_6$ ($\times 10^2$) | $p_7$ ($\times 10^6$) | $T$ range (K) |
|---|---|---|---|---|---|---|---|---|---|
| $K_H$ | $-3.3801$ | $2.495691$ | $0.457081$ | | | $-1.593281$ | $-0.404515$ | $1.541270$ | $273-607$ |
| $K_1$ | $-14.626$ | $-8.204327$ | $-1.402727$ | | | $5.027549$ | $1.268339$ | $-3.879660$ | $273-523$ |
| $K_2$ | $-23.783$ | $-2.484192$ | $-0.748996$ | | | $1.186243$ | $0.389256$ | $-1.297999$ | $273-491$ |
| $K_w$ | $-32.224$ | $-1.481678$ | $8.933802$ | $-2.332199$ | $2.146860$ | | | | $273-323$ |

Constants $p_i$ are for the equation $\ln(K) = p_1 + p_2 T + p_3 T^2 + p_4 T^3 + p_5/T + p_6 \ln(T) + p_7/T^2$.

the activity coefficient of the neutral inorganic species $\mathrm{CO}_{2(\mathrm{aq})}$ is given by

$$\ln(\gamma_{\mathrm{CO}_{2(\mathrm{aq})}}) = \sum_c (2\lambda_{\mathrm{CO}_{2(\mathrm{aq})},c}) m_c + \sum_a (2\lambda_{\mathrm{CO}_{2(\mathrm{aq})},a}) m_a, \tag{20}$$

where $\lambda$ is the specific parameter for the interaction between $\mathrm{CO}_{2(\mathrm{aq})}$ and a cation or an anion. Since ions are present in cation–anion pairs in overall neutral electrolyte solutions, it is impossible to determine the interaction parameter between a single ion and a neutral species in isolation. Therefore, an arbitrary value has to be set for one selected ion. By convention, the value of $\lambda_{\mathrm{CO}_{2(\mathrm{aq})},\mathrm{H}^+}$ is set to zero, forming a reference interaction against which all other ion $\leftrightarrow \mathrm{CO}_{2(\mathrm{aq})}$ interactions in aqueous solution can be calibrated. All $\mathrm{CO}_{2(\mathrm{aq})} \leftrightarrow$ ion interaction parameters used in this study were evaluated by Harvie et al. (1984) and Meng et al. (1995) from solubility data with the values listed in Table 5. When there is no reliable data from the literature, the interaction between the ion of interest and $\mathrm{CO}_{2(\mathrm{aq})}$ is assumed to be zero (i.e., $\lambda_{\mathrm{CO}_{2(\mathrm{aq})},i} = 0.0$). Similarly, as there is almost no data on $\mathrm{CO}_{2(\mathrm{aq})}$ with pure water and organic compounds, it is assumed that the interaction between those species is negligible ($\lambda_{\mathrm{CO}_{2(\mathrm{aq})},\mathrm{org}} = 0.0$).

### 3.3 Reference thermodynamic models

### 3.3.1 Extended Aerosol Inorganics Model (E-AIM)

The Extended Aerosol Inorganics Model (E-AIM), is a thermodynamic model that calculates various equilibria between water and inorganic species. The specific E-AIM subset referenced in this study was developed by Harvie et al. (1984) and Wexler and Clegg (2002) based on Pitzer method, which includes ions: $\mathrm{H}^+$, $\mathrm{NH}_4^+$, $\mathrm{Na}^+$, $\mathrm{K}^+$, $\mathrm{Ca}^{2+}$, $\mathrm{Mg}^{2+}$, $\mathrm{SO}_4^{2-}$, $\mathrm{HSO}_4^-$, $\mathrm{NO}_3^-$, $\mathrm{Cl}^-$, $\mathrm{CO}_3^{2-}$, $\mathrm{HCO}_3^-$, $\mathrm{OH}^-$, and neutral species: $\mathrm{NH}_3$ and $\mathrm{CO}_{2(\mathrm{aq})}$. The equilibria solved in that model cover reactions (R1) to (R5). Due to the scarcity of data involving carbonates and the high validity of E-AIM against various experimental data, we chose this E-AIM model as a benchmark and source of comparison data (in addition to measurements) in this study. The online version of E-AIM is accessible at http://www.aim.env.uea.ac.uk/aim/accent4/main.php (last access: 27 March 2021).





**Table 5.** Summary of $CO_{2(aq)} \leftrightarrow$ ion interaction parameters.

| $i$ | $\lambda_{CO_{2(aq)},i}$ |
|---|---|
| $H^+$ | 0.000 |
| $Li^+$ | 0.000 [a] |
| $Na^+$ | 0.100 |
| $K^+$ | 0.051 |
| $NH_4^+$ | 0.010 |
| $Mg^{2+}$ | 0.183 |
| $Ca^{2+}$ | 0.183 |
| $Cl^-$ | −0.005 |
| $Br^-$ | 0.000 [a] |
| $I^-$ | 0.000 [a] |
| $NO_3^-$ | −0.0457 |
| $IO_3^-$ | 0.000 [a] |
| $OH^-$ | 0.000 |
| $HSO_4^-$ | −0.003 |
| $HCO_3^-$ | 0.000 |
| $SO_4^{2-}$ | 0.097 |
| $CO_3^{2-}$ | 0.000 |

[a] Due to the lack of reliable data on the interaction between the designated ion and $CO_{2(aq)}$, the influence is considered negligible or similar to that of $H^+$ on $CO_{2(aq)}$ in water (i.e., $\lambda_{CO_{2(aq)},i} = 0.000$).

### 3.3.2 Simulating Composition of Atmospheric Particles at Equilibrium (SCAPE 2)

The Simulating Composition of Atmospheric Particles at Equilibrium (SCAPE 2) model is an atmospheric gas–aerosol equilibrium model. It covers ions: $HCO_3^-$, $CO_3^{2-}$, $NH_2CO_2^-$, $H^+$, $Na^+$, $NH_4^+$, $K^+$, $Ca^{2+}$, $Mg^{2+}$, and $CO_{2(aq)}$ (Meng et al., 1995). The two dissociation steps of carbonic acid (R1 and R2) are solved explicitly without the inclusion of water auto-dissociation (R3). The binary and multicomponent activity coefficients of electrolytes in SCAPE 2 are estimated by the Kusik-Meissner method and Pitzer method, respectively. Water activity in a multicomponent system is estimated by the ZSR method based on polynomial fits of available binary water activity data. We refer to these polynomial fits as Meng et al. (1995) data when used as input for the AIOMFAC fit/comparison. Due to the lack of a stated validity range for some of the salts, we only used polynomial fits with defined ones for bicarbonate salts. The complete list of the data types and sources involving carbonate salts and carbonic acid can be found in Table 6. For model-derived data with a notable discrepancy between those from E-AIM and Meng et al. (1995), the weightings of the concerned data were both lowered.





**Table 6.** Data types, number of data points ($N_d$), initial weightings ($w_d$), and references used to fit the MR interaction parameters for cation–anion pairs in the carbonate system.

| Inorganic solutes | Data type [a] | $N_d$ | $w_d^{\text{init}}$ | Reference [b] |
|---|---|---|---|---|
| | Binary mixtures (1 salt/acid + water) | | | |
| $Li_2CO_3$ | $\gamma_\pm$ | 18 | 1.00 | Mamontov and Gorbachev (2020) |
| LiOH | $\gamma_\pm$ | 26 | 1.00 | Hamer and Wu (1972) |
| | $\gamma_\pm$ | 19 | 1.00 | Robinson and Stokes (2002) |
| | $\gamma_\pm$ | 42 | 1.00 | Zaytsev and Aseyev (1992) |
| | $a_w$ | 25 | 1.00 | Hamer and Wu (1972) |
| | $a_w$ | 19 | 1.00 | Robinson and Stokes (2002) |
| | $a_w$ | 22 | 1.00 | Nasirzadeh et al. (2005) |
| $Na_2CO_3$ | $\gamma_\pm$ | 20 | 1.00 | Robinson and Macaskill (1979) |
| | $\gamma_\pm$ | 36 | 1.00 | Goldberg (1981) |
| | $\gamma_\pm$ | 28 | 1.00 | Vanderzee (1982) |
| | $\gamma_\pm$ | 37 | 1.00 | Zaytsev and Aseyev (1992) |
| | $a_w$ | 12 | 1.00 | Peiper and Pitzer (1982) |
| $NaHCO_3$ | $a_w$ | 28 | 1.00 | E-AIM |
| | $a_w$ | 29 | 1.00 | Sarbar et al. (1982a) |
| | $\alpha_{HCO_3^-}$ | 41 | 0.0001 | E-AIM |
| | $\alpha_{HCO_3^-}$ | 2 | 1.00 | Sharygin and Wood (1998) |
| NaOH | $\gamma_\pm$ | 31 | 1.00 | Stokes (1945) |
| | $\gamma_\pm$ | 52 | 1.00 | Hamer and Wu (1972) |
| | $\gamma_\pm$ | 26 | 1.00 | E-AIM |
| | $a_w$ | 31 | 1.00 | Stokes (1945) |
| | $a_w$ | 52 | 1.00 | Hamer and Wu (1972) |
| | $a_w$ | 26 | 1.00 | E-AIM |
| $K_2CO_3$ | $\gamma_\pm$ | 57 | 1.00 | Sarbar et al. (1982b) |
| | $\gamma_\pm$ | 56 | 1.00 | Zaytsev and Aseyev (1992) |
| | $a_w$ | 29 | 1.00 | Roy et al. (1984) |
| $KHCO_3$ [c] | $a_w$ | 17 | 1.00 | Roy et al. (1984) |
| | $a_w$ | 19 | 0.1 | Meng et al. (1995) |
| | $a_w$ | 29 | 0.1 | E-AIM |
| | $\alpha_{HCO_3^-}$ | 29 | 0.0001 | E-AIM |
| KOH | $\gamma_\pm$ | 43 | 1.00 | Hamer and Wu (1972) |
| | $\gamma_\pm$ | 26 | 1.00 | E-AIM |
| | $a_w$ | 43 | 1.00 | Hamer and Wu (1972) |
| | $a_w$ | 26 | 1.00 | E-AIM |
| $(NH_4)_2CO_3$ | $a_w$ | 21 | 1.00 | E-AIM |
| $NH_4HCO_3$ | $a_w$ | 43 | 1.00 | E-AIM |
| | $\alpha_{HCO_3^-}$ | 27 | 0.0001 | E-AIM |
| $MgCO_3$ | $\gamma_\pm$ | 14 | 1.00 | E-AIM |
| | $a_w$ | 14 | 1.00 | E-AIM |
| $Mg(HCO_3)_2$ | $a_w$ | 18 | 5.00 | Meng et al. (1995) |
| $CaCO_3$ | $\gamma_\pm$ | 12 | 1.00 | E-AIM |
| | $a_w$ | 12 | 1.00 | E-AIM |
| $Ca(HCO_3)_2$ | $a_w$ | 28 | 5.00 | Meng et al. (1995) |





| Inorganic solutes | Data type [a] | $N_d$ | $w_d^{\text{init}}$ | Reference [b] |
|---|---|---|---|---|
| Multicomponent mixtures [d] | | | | |
| $Li_2CO_3 + LiCl$ | SLE | 8 | 1.00 | Cheng et al. (2013) |
| $Li_2CO_3 + NaCl$ | SLE | 12 | 1.00 | Cheng et al. (2013) |
| | SLE | 9 | 1.00 | Wang et al. (2018) |
| $Li_2CO_3 + KCl$ | SLE | 8 | 1.00 | Cheng et al. (2013) |
| $Li_2CO_3 + Na_2SO_4$ | SLE | 7 | 1.00 | Wang et al. (2018) |
| $Na_2CO_3 + H_2CO_3 \, [2:1]$ | $a_w$ | 34 | 1.00 | E-AIM |
| $NaCl + NaHCO_3 \, [1:1]$ | $a_w$ | 15 | 1.00 | E-AIM |
| | $\gamma_\pm$ | 21 | 1.00 | Ji et al. (2001) |
| $NaCl + NaHCO_3 \, [9:1]$ | $\gamma_\pm$ | 25 | 1.00 | Ji et al. (2001) |
| $K_2CO_3 + KHCO_3 \, [1:1]$ | $a_w$ | 17 | 1.00 | Roy et al. (1984) |
| $(NH_4)_2CO_3 + Na_2CO_3 \, [1:1]$ | $a_w$ | 16 | 1.00 | E-AIM |
| $MgCO_3 + LiCl$ | SLE | 13 | 1.00 | Dong et al. (2009) |
| $MgCO_3 + NaCl$ | SLE | 13 | 1.00 | Dong et al. (2008) |
| | SLE | 7 | 1.00 | De Visscher et al. (2012) |
| $MgCO_3 + Na_2CO_3$ | SLE | 7 | 1.00 | De Visscher et al. (2012) |
| $MgCO_3 + Na_2SO_4$ | SLE | 8 | 1.00 | De Visscher et al. (2012) |
| $MgCO_3 + MgCl_2$ | SLE | 13 | 1.00 | Dong et al. (2008) |
| $MgCO_3 + LiCl + MgCl_2$ | SLE | 13 | 1.00 | Dong et al. (2009) |
| $MgCO_3 + NaCl + MgCl_2$ | SLE | 13 | 1.00 | Dong et al. (2009) |
| $CaCO_3 + (NH_4)_2SO_4 \, [0.00019:1]$ | $a_w$ | 66 | 1.00 | Onasch et al. (2000) |
| $CaCO_3 + NaCl$ | SLE | 19 | 1.00 | Millero et al. (1984) |

[a] Water activity ($a_w$) data were calculated from osmotic coefficients if not provided in the literature.
[b] Data points from Meng et al. (1995) were generated by using the polynomials in their model parameterization.
[c] As there is major discrepancy between the $a_w$ data, the weighting was reduced.
[d] In the brackets are the corresponding molar ratios of the salts or acids.

## 3.4 Experimental data for AIOMFAC fit

Different data types, including measurements of water activity ($a_w$) or mean molal ion activity coefficients of the electrolytes ($\gamma_\pm$) at known solution compositions, determined phase compositions at vapor–liquid equilibrium (VLE), liquid–liquid equilibrium (LLE) or solid–liquid equilibrium (SLE), are useful for establishing adjustable middle-range parameters in AIOMFAC

(Eqs. 3–5). For water activity data, measurements are usually carried out using macroscopic bulk solutions or microscopic aqueous droplets. For equilibrium (bulk) measurements, assuming water is the only volatile component, its activity on the mole fraction scale can be determined from

$$a_w = \frac{p_w}{p_w^\circ}, \tag{21}$$

with $p_w$ being the water vapor pressure in equilibrium with the solution and $p_w^\circ$ the saturation vapor pressure of pure water at the

measurement temperature. To access the regime beyond saturation of the electrolyte in solvent mixture, an aqueous droplet can be trapped in an electrodynamic balance (EDB) under controlled relative humidity (RH). When a supermicron-sized droplet is





in equilibrium with the gas phase, RH is equivalent to aerosol water activity (i.e., $RH = a_w$). The composition is then evaluated from the droplet's radius or mass change in response to a change in RH relative to the initial dry particle (e.g., Zardini et al., 2008; Krieger et al., 2012). The uncertainty associated with an EDB measurement is generally larger than that of a bulk water

activity measurement due to the combined errors in determining RH and droplet water content reliably from the EDB sensors and raw data conversion (Krieger et al., 2012).

The electromotive force (EMF) method is used to determine the mean activity coefficients of an electrolyte at known concentration. By measuring the electric potential difference between two different electrodes in an electrochemical cell, the mean activity coefficients are derived from the modified Nernst equation and the use of a system-specific thermodynamic model

(e.g., Hamer and Wu, 1972). VLE is either measured under isobaric or isothermal conditions; in the latter case the temperature is typically higher than room temperature (i.e., 298 K). For the fitting of AIOMFAC parameters, and to establish the salt influence on the VLE of the components from the effects of other constituents, the activity coefficient difference of water or organic component $j$ is calculated from data for salt-containing (sc) and salt-free (sf) conditions at the same salt-free mole fraction of $j$ in the solvent mixture as

$$\Delta^{\mathrm{sc,sf}}\gamma_j^{(x)}(x_j') = \gamma_j^{(x),\mathrm{sc}}(x_j') - \gamma_j^{(x),\mathrm{sf}}(x_j').$$
(22)

We use a mixture-specific fit of a Duhem–Margules model (McGlashan, 1963; Soonsin et al., 2010) to compute the composition-dependent $\gamma_j^{(x),\mathrm{sf}}$. The newly fitted parameters are tabulated in Table 7; the complete description of this method is given in Zuend et al. (2011). Experimental LLE data describe the mass or mole fraction phase compositions of coexisting liquid phases, which typically represent a more polar aqueous phase and a less polar organic-rich phase. For our model fit, the measured

composition data are compared with those from the corresponding AIOMFAC-based LLE phase composition predictions, which enable the computation of the cumulative Euclidean distance between the model and measurement points in the phase composition space.

Finally, the SLE data report the liquid phase composition when in equilibrium with a specific solid (crystalline) phase. Under isothermal conditions, the solubility limit of the electrolyte or the organic compound is measured for different mixing

compositions. At equilibrium, the liquid phase activity of the organic (if it is the solid) or the molal ion activity product of the electrolyte (for a salt as solid), should be at a constant value (for constant temperature), while the liquid phase concentration may vary (for ternary and higher mixtures). Therefore, the use of SLE data involves the comparison of the solution mass fractions predicted by AIOMFAC after solving for SLE with those from the measurements.

### 3.5 Alternative methods for the determination of interaction parameters

Due to the general lack of experimental data for mixtures of iodide, iodate, or carbonate electrolytes with organic compounds, alternative methods were adopted for the determination of AIOMFAC interaction parameters between those ions and the organic main groups. A linear regression analysis is chosen to have a broader coverage of the interaction parameters for $I^-$ and organic compounds. Based on the similarities in physical properties and interaction parameter patterns of other ions in





**Table 7.** Coefficients for the Duhem–Margules excess Gibbs energy model (McGlashan, 1963; Soonsin et al., 2010) fitted to salt-free binary organic solvent mixture data.

| Binary system components | $p$ (kPa) | $T$ (K) | Coefficients of the model fit | | | | Exp. data |
|---|---|---|---|---|---|---|---|
| | | | $c_1$ $(\times 10^{-1})$ | $c_2$ $(\times 10^{-3})$ | $c_3$ $(\times 10^{-1})$ | $c_4$ $(\times 10^{-2})$ | |
| methanol(1) + benzene(2) | 101.3 | 331–344 | 18.601 | −183.47 | 2.1440 | −2.3280 | a |
| methanol(1) + dioxane(2) | 101.3 | 338–350 | 8.2870 | −0.3491 | 1.8199 | −19.396 | b |
| acetone(1) + methanol(2) | 101.3, 99.8–101.2 | 328–337 | 5.8277 | 17.507 | −0.2556 | 0.1554 | c |

Coefficients and model expressions are as those of Soonsin et al. (2010); see their Appendix A.
Experimental data sources: (a) reference Hiaki and Kawai (1999); Li et al. (2019), (b) reference Liu et al. (1998), (c) reference Iliuta and Thyrion (1995); Chen et al. (2015); Tu et al. (1997); Li et al. (2014).

the model, the interaction parameters for $Br^-$ ↔ organic main group and $Mg^{2+}$ ↔ organic main group are selected as the
independent variables for the analysis (i.e., to predict pertinent parameters for $I^-$). Parameters for organic main group $CH_n$, $CH_n^{[OH]}$, OH and COOH with $Br^-$ and $Mg^{2+}$ are used to determine and/or validate parameters for the same organic main groups with $I^-$. The relationship is then used to estimate the interaction parameters for $I^-$ and other organic main groups that lack experimental data. The regression is performed separately for the $b_{k,i}^{(1)}$ and $b_{k,i}^{(2)}$ values, with additional comparison of the combined $B_{k,i}$ values (Eq. 3) using binary solutions of the following ionic strengths: 0.001, 0.1, 10, and 100 $mol\,kg^{-1}$, which
cover a broad range of electrolyte concentrations.

Because there is no thermodynamic data for the mixture of iodate electrolytes with organic compounds, we make the crude estimation of the interaction parameters for $IO_3^-$ ↔ organic main group based on the comparison of water activity or mean molal activity coefficients in binary iodate and other electrolyte solutions. The selection of the replacement anion is based on the similarities in thermodynamic data and ions' physical properties.

Since the main type of experimental data covering the mixture of carbonate salts + water + organic compounds is LLE data, the associated relatively large uncertainty and the limited number of data sources make it unfeasible for the optimization method to determine parameters in a reasonable value range (based on experience for other salts). Instead, we estimate the interaction parameters for carbonate ions and organic compounds based on those for sulfate ions due to the similar ion size and electric charge:

$$CO_3^{2-} \leftrightarrow \text{ organic main group } \approx SO_4^{2-} \leftrightarrow \text{ organic main group,} \tag{23}$$

$$HCO_3^- \leftrightarrow \text{ organic main group } \approx HSO_4^- \leftrightarrow \text{ organic main group.} \tag{24}$$

In short, methods like those described in this section can be adopted as a first-order estimation approach for interactions lacking support by high-quality experimental data for a more sophisticated model parameter determination. The updated AIOMFAC group interaction matrix indicating all available binary interactions in the model is shown in Fig 1.



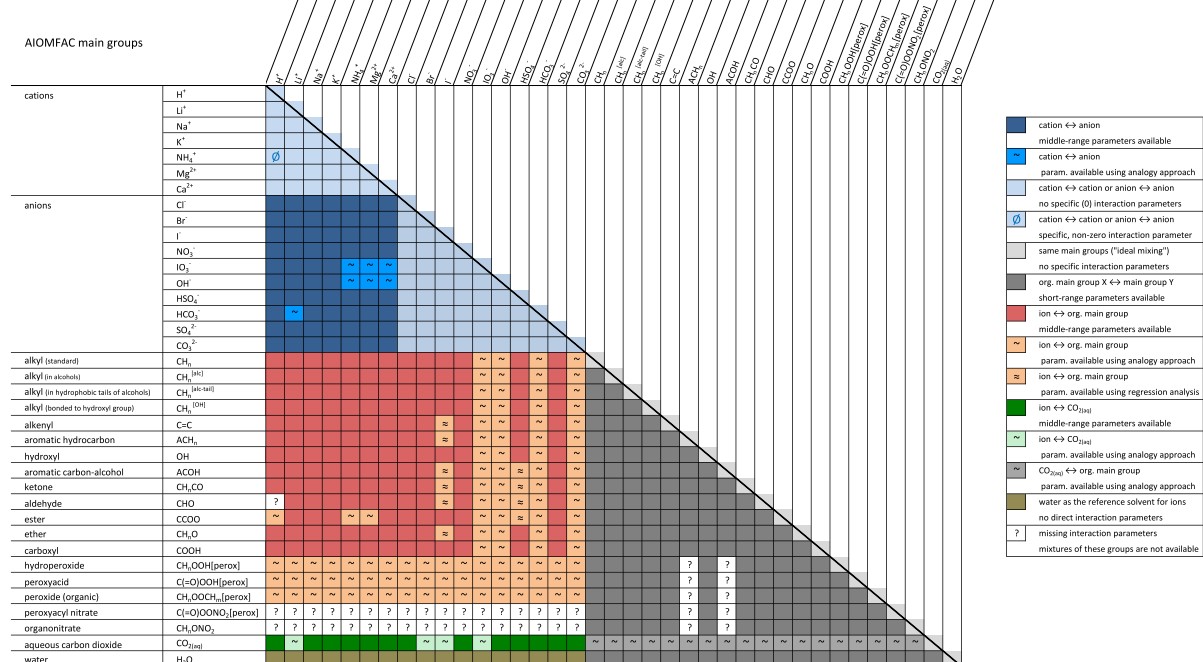

**Figure 1.** Revised AIOMFAC interaction matrix indicating available and missing binary interaction parameters. Parameters available based on regression analysis are depicted by symbol ($\approx$), those based on substitution are depicted by symbol ($\sim$). Figure adapted based on original Fig. 4 from https://aiomfac.lab.mcgill.ca/about.html.

## 4 Results and Discussion

### 4.1 Aqueous inorganic iodine electrolytes

Using the outlined optimization procedure in Sect. 2.2, we determined the five parameters $b_{c,a}^{(1)}$, $b_{c,a}^{(2)}$, $b_{c,a}^{(3)}$, $c_{c,a}^{(1)}$, and $c_{c,a}^{(2)}$ describing the middle-range interaction between cations and anions in water; the fitted parameter sets are listed in Table 8. Here, we included the inorganic iodide electrolytes associated with all cations in the model, namely NaI, KI, HI, LiI, NH$_4$I, MgI$_2$, and CaI$_2$. Figure 2 shows a model–measurement comparison of the various binary aqueous iodide solutions. It is clear that AIOMFAC is able to fully capture the mixing behavior in the measurement range. Beyond the concentration range covered by the measurements, our model is able to make physically reasonable extrapolations of solution water activity to zero water content. Panels (h) and (i) of Fig. 2 show the comparison of predicted water activity and mean molal activity coefficient curves for all iodide systems. The dashed diagonal line denotes the water activity in an ideal mixture. Electrolytes like LiI, HI, CaI$_2$, and MgI$_2$ deviate more from ideal mixing behavior than NH$_4$I and KI. This pattern has been observed in chloride, nitrate and other electrolyte series as well (Zuend et al., 2008). A likely reason is that the smaller and/or divalent cations Li$^+$, Mg$^{2+}$ and Ca$^{2+}$ contain a higher relative surface charge density than NH$_4^+$ and K$^+$ (Zuend et al., 2008). While error bars are not reported





**Table 8.** Determined middle-range interaction parameters between cation–anion pairs for iodide and iodate electrolytes [a].

| $c$ | $a$ | $b_{c,a}^{(1)}$ (kg mol$^{-1}$) | $b_{c,a}^{(2)}$ (kg mol$^{-1}$) | $b_{c,a}^{(3)}$ (kg$^{1/2}$ mol$^{-1/2}$) | $c_{c,a}^{(1)}$ (kg$^2$ mol$^{-2}$) | $c_{c,a}^{(2)}$ (kg$^{1/2}$ mol$^{-1/2}$) |
|---|---|---|---|---|---|---|
| $H^+$ | $I^-$ | 0.509103 | 0.179367 | 0.8 | $-0.099439$ | 1.371292 |
| $Li^+$ | $I^-$ | 0.185889 | 0.608090 | 0.8 | 0.041814 | 0.413415 |
| $Na^+$ | $I^-$ | 0.068350 | 0.379661 | 0.8 | 0.041454 | 0.499456 |
| $K^+$ | $I^-$ | 0.054421 | 0.188226 | 0.8 | 0.025346 | 0.660773 |
| $NH_4^+$ | $I^-$ | 0.092192 | $-0.046784$ | 0.8 | 0.081996 | 1.324042 |
| $Mg^{2+}$ | $I^-$ | 0.961297 | 0.773564 | 0.8 | $-0.472857$ | 1.082782 |
| $Ca^{2+}$ | $I^-$ | 0.759764 | 1.103194 | 0.8 | $-0.672438$ | 1.518045 |
| $H^+$ | $IO_3^-$ | 0.014981 | $-3.206355$ | 0.8 | $-0.042731$ | 0.535963 |
| $Li^+$ | $IO_3^-$ | 0.076050 | $-0.249305$ | 0.296890 | $-0.128529$ | 1.742383 |
| $Na^+$ | $IO_3^-$ | 0.013438 | $-0.964866$ | 1.304777 | $-0.137069$ | 1.362620 |
| $K^+$ | $IO_3^-$ | 0.012532 | $-1.839558$ | 1.274413 | $-1.461885$ | 1.361224 |
| $NH_4^{+\ b}$ | $IO_3^-$ | $-0.000057$ | $-0.171746$ | 0.260000 | 0.005510 | 0.529762 |
| $Mg^{2+}$ | $IO_3^-$ | 0.121261 | 0.171472 | 0.8 | $-0.532150$ | 0.877292 |
| $Ca^{2+\ b}$ | $IO_3^-$ | 0.163282 | 0.203681 | 0.8 | $-0.075452$ | 1.210906 |

[a] The number of digits listed reflects the approximate precision used in the model code and does not imply that all digits are significant figures.
[b] Estimated from the interaction parameters for designated cations $\leftrightarrow NO_3^-$.

for most experimental data sources, based on the good agreement between different experimental datasets, it is safe to assume that the measurement errors are within the range of the symbol size.

Given the lack of reliable aqueous iodate solution data, especially at the concentrated range, aqueous $NaIO_3$, $KIO_3$, $LiIO_3$, $HIO_3$, and $Mg(IO_3)_2$ solutions were implemented based on experimental data in AIOMFAC in this model extension. Aqueous $NH_4IO_3$ and $Ca(IO_3)_2$ solutions were added based on an alternative method instead as discussed in Sect. 4.4. Figure 3 shows the experimental data and corresponding AIOMFAC predictions of water activity and the mean molal activity coefficients of $NaIO_3$, $KIO_3$, $LiIO_3$, and $HIO_3$ electrolytes at room temperature. The agreement between the experiments and the model

predictions is very good, except for one data point at the solubility limit of $KIO_3$ reported by Apelblat and Korin (1998) and part of the $a_w$ measurements of $HIO_3$ solutions. Since our own measurements of the $KIO_3$ water activity close to the solubility limit at 293 K is considered to be of high quality (see SI Sect. S4), it is very likely that the reported water activity value by Apelblat and Korin (1998) is erroneous. Given $HIO_3$ water activity data reported by Goldman et al. (1974) agree quite well with our own measurements (see SI Sect. S4), the weightings of the bulk measurements by Kumar et al. (2010) and the

subsequent EDB measurements by Murray et al. (2012) were lowered during the parameter fitting process. The reason for such disagreement between different datasets is unclear and the focus of this work is not to explain the discrepancies. Additionally, due to the low solubility of $NaIO_3$ and $KIO_3$ in water, the validity of the AIOMFAC predictions of binary aqueous $NaIO_3$ and $KIO_3$ solutions may be compromised at high ionic strength (e.g., in supersaturated salt solutions) due to a lack of data for model fit/validation.





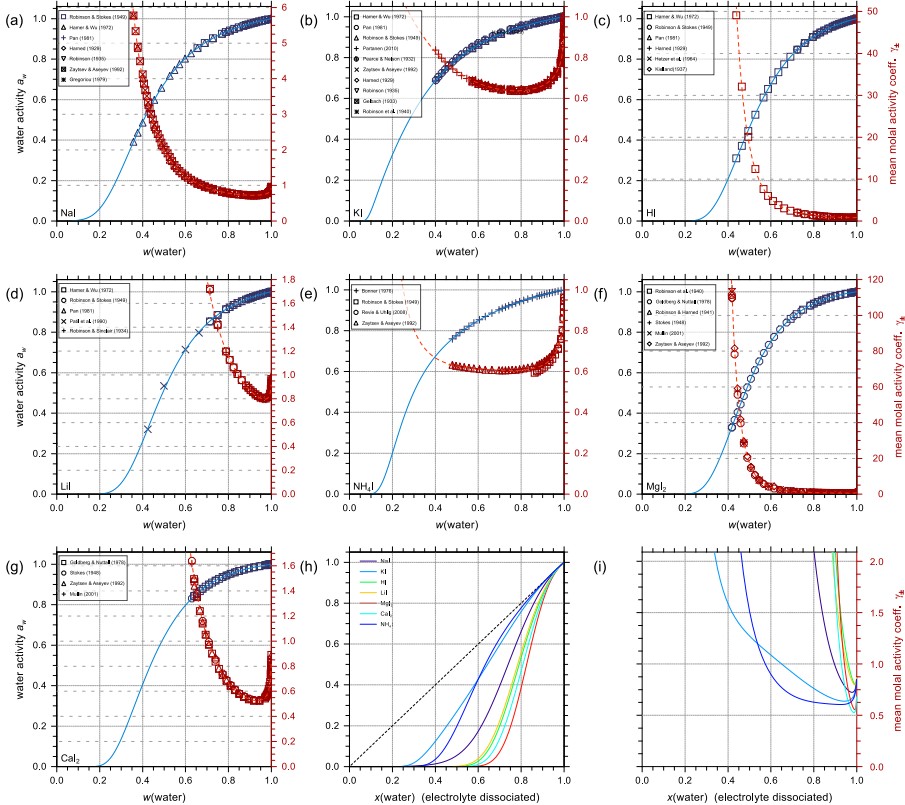

**Figure 2.** Water activities and mean molal activity coefficients of the electrolytes in binary aqueous iodide solutions near 298.15 K. The symbols represent the experimental data and the solid curves show the AIOMFAC predictions. The panels **(a)** to **(g)** show as x-axis the mass fraction of water, $w$(water), while panels **(h)** and **(i)** show the mixture composition in terms of $x$(water), the mole fraction of water defined with respect to dissociated electrolytes. The left y-axis (blue curves, symbols) indicates water activity and the right y-axis (red curves, symbols) indicates the mean molal activity coefficients. Electrolytes: **(a)** NaI, **(b)** KI, **(c)** HI, **(d)** LiI, **(e)** NH$_4$I, **(f)** MgI$_2$, **(g)** CaI$_2$. **(h)** AIOMFAC-calculated water activity curves of all the iodide electrolytes implemented; the dashed black line indicates the water activity of an ideal mixture. **(i)** corresponding mean molal activity coefficient curves.

## 4.2 Parameterization of iodide salt interactions with organic compounds

The resulting parameters of the model optimization for I$^-$ ↔ organic main groups interactions (to calculate $B_{k,i}$), using two different methods, are provided in Table 9. Since there are many data types and sources involved, we will focus on the discussion of a selection of systems containing carboxylic acids and alcohols in the following. Plots depicting some additional systems are provided in Fig. S1.

### 4.2.1 Iodide–organic acid interactions

Organic compounds such as carboxylic acids contribute a large fraction of the water-soluble organic aerosol mass, including in marine environments (e.g., Saxena and Hildemann, 1996; Decesari et al., 2000). It is thus important to accurately predict





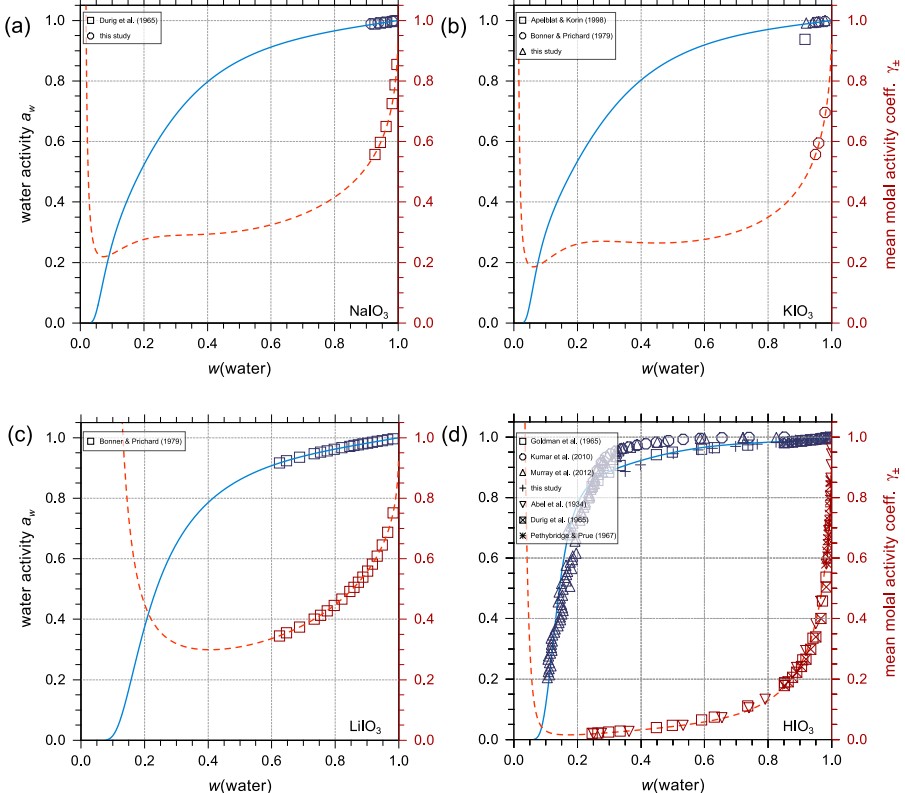

**Figure 3.** Water activities and mean molal ion activity coefficients of the electrolytes in binary aqueous iodate solutions at room temperature. Symbols: experimental data (see legend); solid blue and dashed red curves: fitted AIOMFAC model. **(a)** $NaIO_3$, **(b)** $KIO_3$, **(c)** $LiIO_3$, **(d)** $HIO_3$.

the activity coefficients in iodide electrolytes + organic acid mixtures and to evaluate the potential for liquid–liquid phase separation. Figure 4 shows experimental water activity data and AIOMFAC predictions of ternary systems NaI + water +

different carboxylic acids: malonic acid (panel a), citric acid (panel b) and glutaric acid (panels c and d). The mass ratio between NaI and the organic acid in each set of the experiments is kept constant, allowing for a direct comparison with the shown model curves. The dashed curves show the water activities of the corresponding salt-free solvent mixture to highlight the effect of adding salt to the solution. It is this difference between salt-free and salt-containing solutions that should be explained by the interactions among ions with water and the organic functional groups present (at least within AIOMFAC). In general,

the model–measurement agreement is good especially at high water content ($a_w > 0.8$). In the case of water + citric acid + NaI, AIOMFAC slightly overpredicts the water activity compared to the EDB measurements for $a_w < 0.8$. This deviation is largely attributed to AIOMFAC's group–contribution approach with further discussion provided in Sect. 4.2.3.



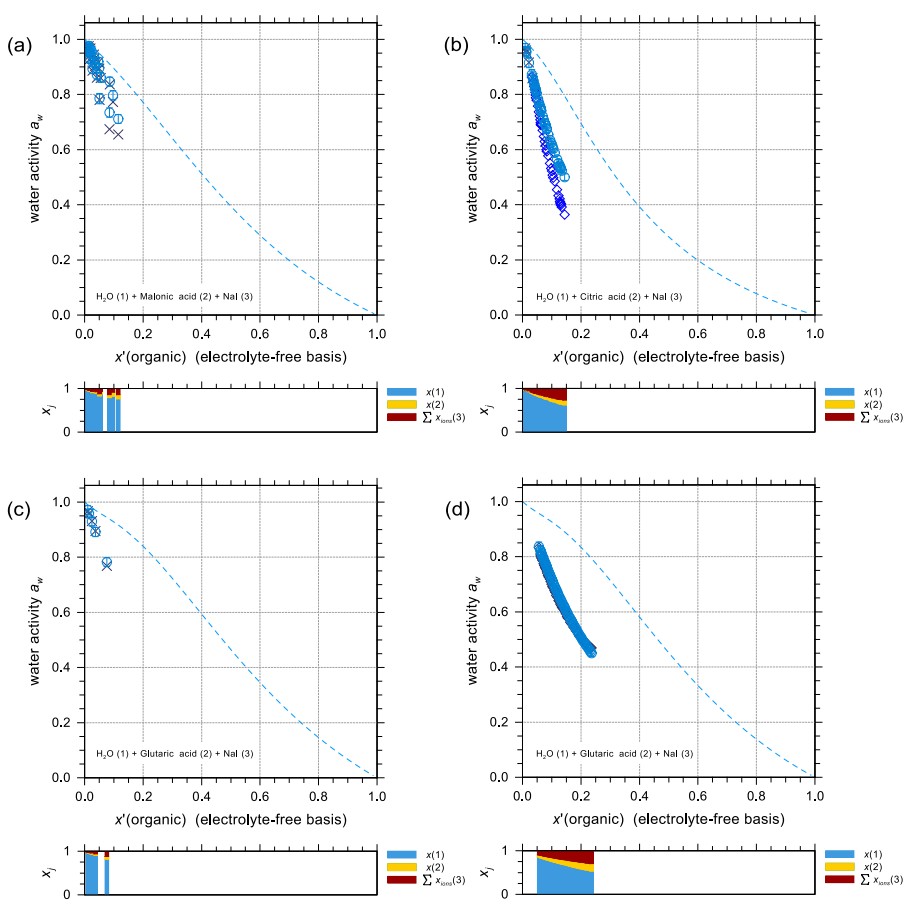

**Figure 4.** Water activities of NaI + water + carboxylic acid systems. Symbols: (×) bulk solution measurements at 298.15 K; (◇) EDB measurements at various temperatures (see details in SI Sect. S4); (○) AIOMFAC predictions at the corresponding experimental temperatures including estimated model sensitivity shown as error bars. **(a)** Water (1) + malonic acid (2) + NaI (3); experiments at various mixing ratios (this study). **(b)** Water (1) + citric acid (2) + NaI (3); experiments at 1 : 1 mass ratio of citric acid : NaI (this study). **(c)** and **(d)** Water (1) + glutaric acid (2) + NaI (3); experiments at 1 : 1 mass ratio of glutaric acid : NaI (this study). The dashed curves indicate the water activities of the corresponding salt-free systems. The composition bar panels show the stacked mole fractions of the three components with respect to dissociated NaI.





**Table 9.** Determined middle-range interaction parameters $b_{k,i}^{(1)}$ (kg mol$^{-1}$) and $b_{k,i}^{(2)}$ (kg mol$^{-1}$) between I$^-$ and organic main groups. The values were either determined from a AIOMFAC model fit to experimental data or by using a regression method to predict the values based on the known parameters of other ions interacting with the listed organic main groups [a].

| | Fitted from experimental data [b] | | Evaluated from regression analysis | |
|---|---|---|---|---|
| | $b_{k,i}^{(1)}$ | $b_{k,i}^{(2)}$ | $b_{k,i}^{(1)}$ | $b_{k,i}^{(2)}$ |
| CH$_n$ | $4.87893\times10^{-2}$ | $-8.51607\times10^{-3}$ | – | – |
| CH$_n^{[OH]}$ | $3.42781\times10^{-2}$ | $-6.08293\times10^{-3}$ | – | – |
| OH | $1.93902\times10^{-2}$ | $-1.63692\times10^{-2}$ | – | – |
| COOH | $3.42783\times10^{-2}$ | $-3.64764\times10^{-2}$ | – | – |
| CH$_n$CO | $7.55114\times10^{-2}$ | $-2.11346\times10^{-2}$ | $7.37680\times10^{-2}$ | $-1.52019\times10^{-2}$ |
| CHO | – | – | $3.89605\times10^{-2}$ | $-2.96326\times10^{-2}$ |
| CH$_n$O | $4.87893\times10^{-2}$ | $-8.51607\times10^{-3}$ | $4.11765\times10^{-2}$ | $-1.41737\times10^{-2}$ |
| CCOO | $3.42779\times10^{-2}$ | $2.16934\times10^{-2}$ | – | – |
| C=C | – | – | $7.13425\times10^{-2}$ | $-1.30138\times10^{-2}$ |
| ACH$_n$ | $3.42781\times10^{-2}$ | $-6.08291\times10^{-3}$ | $4.26000\times10^{-2}$ | $-1.89659\times10^{-2}$ |
| ACOH | $3.51122\times10^{-2}$ | $-1.49678\times10^{-2}$ | $4.13342\times10^{-2}$ | $-2.43409\times10^{-2}$ |

[a] The number of digits listed reflects the approximate precision used in the model code and does not imply that all digits are significant figures.
[b] Parameters in black font are used in the revised AIOMFAC model while those in gray are shown here for comparison/validation purposes only.

### 4.2.2 Iodide–alcohol/polyol interactions

Figure 5 shows a selection of vapor–liquid equilibrium (VLE), water activity ($a_w$), liquid–liquid equilibrium (LLE) and sol-
ubility (SLE) experimental data and pertaining model calculations for iodide salt + water + alcohol mixtures. Figure 5a is
VLE system of water + ethanol + KI under isobaric condition in the temperature range from 351 to 366 K. The salting-out
effect on the organic (large positive activity coefficient deviation of ethanol) is most prominent for small mole fractions of
ethanol. Note that the scatter among data points in such figures is also due to measurements of ternary solutions of different
salt contents while at similar electrolyte-free mole fractions of the organic. Surprisingly, experimental data show some strong
salting-out behavior on water in water-rich regimes which may come from experimental uncertainties and/or AIOMFAC fits
not being perfect. The water activity measurements and AIOMFAC predictions for the ternary system water + sorbitol + NaI
at 288 K are shown in Fig. 5b. The mass ratio of sorbitol to NaI is kept as 1 : 1 in the experiments. AIOMFAC predictions are
substantially lower, by about 0.1 units, than EDB measurements at lower water activity, while they are much better at higher
water contents. Figure 5c shows a LLE phase diagram of water + 1-butanol + KI at 298.15 K, where the compositions of the
two coexisting phases are connected by tie-lines. Since the potential errors associated with this type of measurements are high,
even with slight deviations, AIOMFAC is considered to be in very good agreement with the experiments. Figure 5d shows
the solubility limit of KI for different mixing ratios of water + 2-propanol at 298.15 K. In the water-rich composition range,
AIOMFAC underpredicts the salt solubilities slightly, while it overpredicts solubility when the system enters the organic-rich



regime, e.g., for $x'(\text{water}) < 0.6$. The different directions of model deviations are a typical result of the optimization trade-off
when using a group–contribution approach to simultaneously represent many compounds/systems. Since we have a rather big
data collection (28 sets in total) for systems covering $I^-$ + organic acids and $I^-$ + aliphatic alcohols, the achieved fitting
quality of the two targeted sets of parameters, $I^- \leftrightarrow \text{COOH}$ and $I^- \leftrightarrow \text{OH}$, is considered high. In addition, the fitting quality
of interaction parameters for $I^- \leftrightarrow \text{CH}_n^{[\text{OH}]}$ and $I^- \leftrightarrow \text{CH}_n$ may benefit from this diversity in datasets. However, the validity
of other parameters (covering a range of other organic main groups) may be limited since the measurement data either only
covers very dilute concentration ranges or the number of datasets is small. Hence, this motivated the adoption of alternative
approaches for determining interaction parameters for iodide interactions with such organic main groups. The results from
those approaches are further discussed in Sect. 4.4.

### 4.2.3 Discussion of uncertainties in determining $I^-$ interaction parameters with organic main groups

The determination of interactions for $I^- \leftrightarrow$ organic main groups is based on the previously known parameters between cations
and organic main groups from Zuend et al. (2011) and those between $I^-$ and other cations from this study. Unlike binary
solutions involving inorganic electrolytes and water, for which the AIOMFAC calculations of water activity and mean molal
activity coefficients are in excellent agreement with the experimental findings, there is typically notable model–measurement
deviations when the group–contribution approach is applied to organic–inorganic systems, including the systems involving
iodide ions. Several reasons could explain the above phenomenon. First, uncertainties already exist in the parameterization
of mixtures of water and organic compounds, represented by the UNIFAC-based group–contribution mixing model within
AIOMFAC (Zuend et al., 2011; Ganbavale et al., 2015). Hence, an observed model–measurement deviation in a certain iodide-
containing mixture should not be expected to be explained solely (or completely) by the interactions among $I^-$ and organic
main groups (i.e., the parameters to be determined via optimization). Since most measurements available and used in this
study involve either NaI or KI as iodide salts, the $I^- \leftrightarrow$ organic main group interactions of interest become potentially biased
by the involved counterions ($\text{Na}^+$, $\text{K}^+$). This is the case because the parameters of those cations with organic main groups
typically may also be affected by a degree of uncertainty, which could be systematic, i.e., dependent on ionic strength and
organic main group amounts. However, given that a large number of datasets were involved in optimizing the interactions
of both $\text{Na}^+$ and $\text{K}^+$ with organic main groups in past work (Zuend et al., 2011), the bias from the use of these cations
together with $I^-$ should be relatively small. In addition, considering the same class of organic compounds (e.g., carboxylic
acids), the model–measurement agreement for some systems is sufficiently good (e.g., glutaric acid + NaI), yet clearly less
than optimal for other systems (e.g., citric acid + NaI). This is largely a result of imperfection (or lack of specificity) in
AIOMFAC's group–contribution approach when applied to certain organic compounds, since the model has not been fitted to
perform optimally for each specific system. While system-specific parameterizations of AIOMFAC are possible, likely leading
to improvements in problematic cases, such specificity is not the goal of AIOMFAC, since one would lose the predictability
feature of the group–contribution model, which is clearly of importance for applications to atmospheric aerosol systems. Thus,
the deviations between the presented model results and the experiments should not be solely attributed to the $I^- \leftrightarrow$ main
group interactions. If one were to fit AIOMFAC only to achieve better agreement with a subset of experiments, issues of



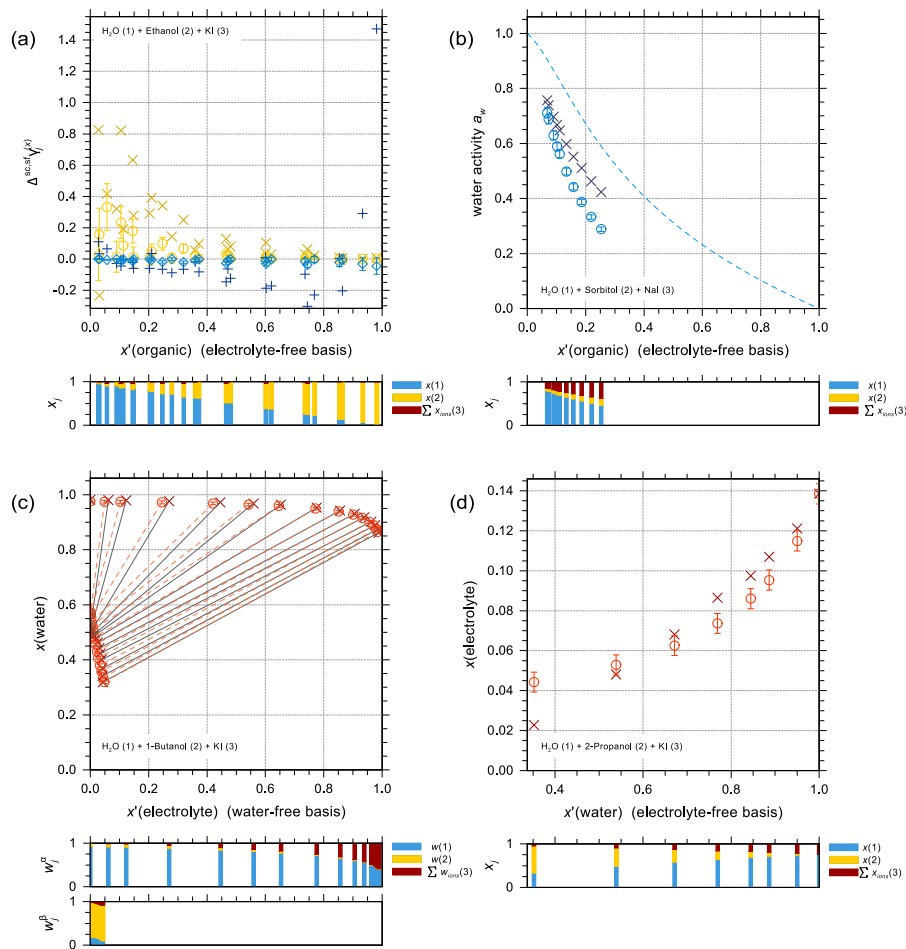

**Figure 5.** Different types of experimental data ($\times$, $+$) and AIOMFAC predictions ($\bigcirc$, $\diamond$) for water $+$ alcohol $+$ iodide salt systems with model sensitivity indicated by error bars. **(a)** VLE of water (1) $+$ ethanol (2) $+$ KI (3) at 351–366 K; experiments by Chen and Zhang (2003). **(b)** Water activity of water (1) $+$ sorbitol (2) $+$ NaI (3) at 288 K; experimental data (this study) for 1 : 1 mass ratio of sorbitol : NaI. **(c)** LLE of water (1) $+$ 1-butanol (2) $+$ KI (3) at 298 K; experiments by Al-Sahhaf and Kapetanovic (1997). **(d)** SLE of water (1) $+$ 2-propanol (2) $+$ KI (3) at 298 K; experiments by Pawar et al. (2012). The composition bar graphs show the mass fractions (in c) or the mole fractions (a, b, d) of the components with respect to dissociated salts.





overfitting or unbalanced parameter constraints would be expected. The group–contribution approach and complexity of the underlying AIOMFAC expressions present a trade-off between a wider representation of organic compounds using a limited

set of parameters and higher accuracy in specific systems. Ideally, the database of measurements used for the optimization of model parameters would include thousands of datasets covering a wide range of organic compounds (thereby many main groups in different ratios), different counterions, mixture compositions from dilute to highly concentrated, a broad range of temperatures, and small standard deviations in measurements. At present, the situation for constraining inorganic ion $\leftrightarrow$ organic main group interactions is quite far from the outlined ideal one.

## 4.3 Aqueous inorganic carbonate electrolytes


The complete set of middle-range interaction parameters involving carbonic acid and salts are listed in Table 10. Some $OH^-$ electrolytes are poorly soluble in water; therefore, in the case of difficulty in fitting $Ca^{2+} \leftrightarrow OH^-$ and $Mg^{2+} \leftrightarrow OH^-$ interactions due to insufficient data coverage over a range of water activities, the relevant parameters were estimated based on those of cation $\leftrightarrow I^-$ (analogy approach). Since the concentration of $OH^-$ is generally very small compared to other ions in an

aqueous solution, such an estimation is justifiable. In the following, we first discuss results for carbonate salts ($CO_3^{2-}$) and then highlight mixing behavior in $H_2CO_3$ and $NaHCO_3$ solutions. In addition, the results for selected bicarbonate salts ($HCO_3^-$) are discussed to show the model's capability in solving the coupled equilibria.

Analogous to the treatment of iodine salts, all carbonate salts are assumed to be completely dissociated (in the absence of significant amounts of bicarbonate). As shown in Fig. 6, AIOMFAC predictions are in excellent agreement with either

experimental data or E-AIM predictions (generated data within the applicable range of that model). Using such data, we established the set of interaction parameters for different cations and the anion $CO_3^{2-}$ in the absence of bicarbonate and other ions. This enables a level of separation in terms of quantifying the $CO_3^{2-}$ influence prior to conducting the AIOMFAC parameter estimation for mixtures containing substantial amounts of the other ions of the bicarbonate/carbonic acid system, like $H^+$ and $HCO_3^-$.

## 4.3.1 Closed-system scenario


The top panel of Fig. 7 shows AIOMFAC predictions of water activity and corresponding $HCO_3^-$ degree of dissociation in aqueous $H_2CO_3$ and $NaHCO_3$ solutions. The bottom panel is the pH and partial pressure of $CO_{2(g)}$ of the same system as in the panels above them, plotted against mass fraction of water. In the case of $H_2CO_3$, both water activity and the dissociation degree data agree very well among the two models (E-AIM and AIOMFAC). The degree of $HCO_3^-$ dissociation is close to

1.0 except for extreme dilution in water, meaning presence of a low ratio of $HCO_3^-$ after the system reaches equilibrium. The composition bar panels indicate that the (closed, liquid) system is mainly composed of $H_2O$ ($x(1)$) and $CO_{2(aq)}$ ($x(2)$) with seemingly negligible amounts of ions across the whole concentration range. This phenomenon validates the assumption that given a lack of experimental data, assuming negligible interactions between water and $CO_{2(aq)}$ (see Sect. 3.2.1) is accurate. It is also noticeable that the water activity of an aqueous $H_2CO_3$ solution never reaches $a_w$ values below 0.6. This is the case

because of the involved mass balance/exchange among all the solute species during the dissociation process. The backward





**Table 10.** Determined middle-range interaction parameters between cation–anion pairs in aqueous carbonate electrolytes [a].

| $c$ | $a$ | $b_{c,a}^{(1)}$ (kg mol$^{-1}$) | $b_{c,a}^{(2)}$ (kg mol$^{-1}$) | $b_{c,a}^{(3)}$ (kg$^{1/2}$ mol$^{-1/2}$) | $c_{c,a}^{(1)}$ (kg$^2$ mol$^{-2}$) | $c_{c,a}^{(2)}$ (kg$^{1/2}$ mol$^{-1/2}$) |
|---|---|---|---|---|---|---|
| $H^+$ | $CO_3^{2-}$ | −0.152205 | 0.170201 | 0.8 | 0.086856 | 0.596222 |
| $H^+$ | $HCO_3^-$ | 0.048650 | −0.473652 | 0.8 | 0.009909 | 0.328049 |
| $H^+$ | $OH^-$ | 0.0 | 0.0 | 0.0 | 0.0 | 0.0 |
| $Li^+$ | $CO_3^{2-}$ | 0.000752 | −1.604524 | 0.8 | −0.009512 | 0.365258 |
| $Li^{+\ b}$ | $HCO_3^-$ | 0.092903 | −0.882375 | 0.565523 | 0.005925 | 0.624926 |
| $Li^+$ | $OH^-$ | 0.075665 | −0.913306 | 0.8 | 0.691057 | 2.347786 |
| $Na^+$ | $CO_3^{2-}$ | 0.106958 | −0.384657 | 0.8 | −0.198184 | 0.846353 |
| $Na^+$ | $HCO_3^-$ | 0.064999 | −0.075189 | 0.8 | −0.057300 | 0.932761 |
| $Na^+$ | $OH^-$ | 0.023840 | 0.669563 | 0.226869 | −0.523049 | 1.303755 |
| $K^+$ | $CO_3^{2-}$ | 0.192681 | 0.135478 | 0.8 | −0.181906 | 0.943337 |
| $K^+$ | $HCO_3^-$ | 0.092903 | −0.882375 | 0.565523 | 0.005925 | 0.624926 |
| $K^+$ | $OH^-$ | 0.251315 | 0.536471 | 0.8 | −0.759101 | 1.936355 |
| $NH_4^+$ | $CO_3^{2-}$ | 0.007832 | −2.711595 | 0.8 | −0.072644 | 0.536251 |
| $NH_4^+$ | $HCO_3^-$ | 0.171403 | −0.210465 | 0.435309 | −0.045620 | 0.580235 |
| $NH_4^{+\ b}$ | $OH^-$ | 0.251315 | 0.536471 | 0.8 | −0.759101 | 1.936355 |
| $Mg^{2+}$ | $CO_3^{2-}$ | 0.686152 | −2.238171 | 0.105783 | −0.565391 | 1.200405 |
| $Mg^{2+}$ | $HCO_3^-$ | 0.187740 | −1.024355 | 0.183037 | 0.216936 | 1.396086 |
| $Mg^{2+\ c}$ | $OH^-$ | 0.961297 | 0.773564 | 0.8 | −0.472857 | 1.082782 |
| $Ca^{2+}$ | $CO_3^{2-}$ | 0.603777 | −0.008043 | 1.025727 | −0.499894 | 1.589443 |
| $Ca^{2+}$ | $HCO_3^-$ | 0.401202 | −0.795623 | 0.8 | −0.644270 | 1.088639 |
| $Ca^{2+\ c}$ | $OH^-$ | 0.759764 | 1.103194 | 0.8 | −0.672438 | 1.518045 |

[a] The number of digits listed reflects the approximate precision used in the model code and does not imply that all digits are significant figures.
[b] Estimated from the interaction parameters for $K^+$ ↔ designated anions.
[c] Estimated from the interaction parameters for for designated cations ↔ $I^-$.

reaction of (R1) makes the amount of $CO_{2(aq)}$ at high acid input comparable to that of water, which lowers the ionic strength consequently at equilibrium. Because the activity coefficient and activity are parameterized as functions of ionic strength in AIOMFAC, the low concentration of ionic species at equilibrium with initially high acid input makes low water activity in such systems unreachable. Moreover, even as a weak acid, the pH of the carbonic acid system remains below 4.0 until very high
dilution in water. This outcome could only occur under the assumption that all the species remain in a single aqueous phase. However, when the (equilibrium) partial pressure of $CO_{2(g)}$ becomes extremely high (i.e., $p_{CO_{2(g)}} > 500 \, \mathrm{ppm_v}$, parts per million by volume), the single phase scenario of a closed liquid system may not occur under normal atmospheric conditions where $p_{CO_{2(g)}}$ is near 400 ppm$_v$. Hence, when the predicted $CO_{2(g)}$ partial pressure is higher than the typical ambient atmospheric value of about 415 ppm$_v$ presently in 2021 (Dlugokencky and Tans, 2021), this would be an indication that a potentially large
fraction of the dissolved $CO_{2(aq)}$ would partition to the gas phase, if allowed (i.e., if treated as an open system) as discussed in Sect. 4.3.2.





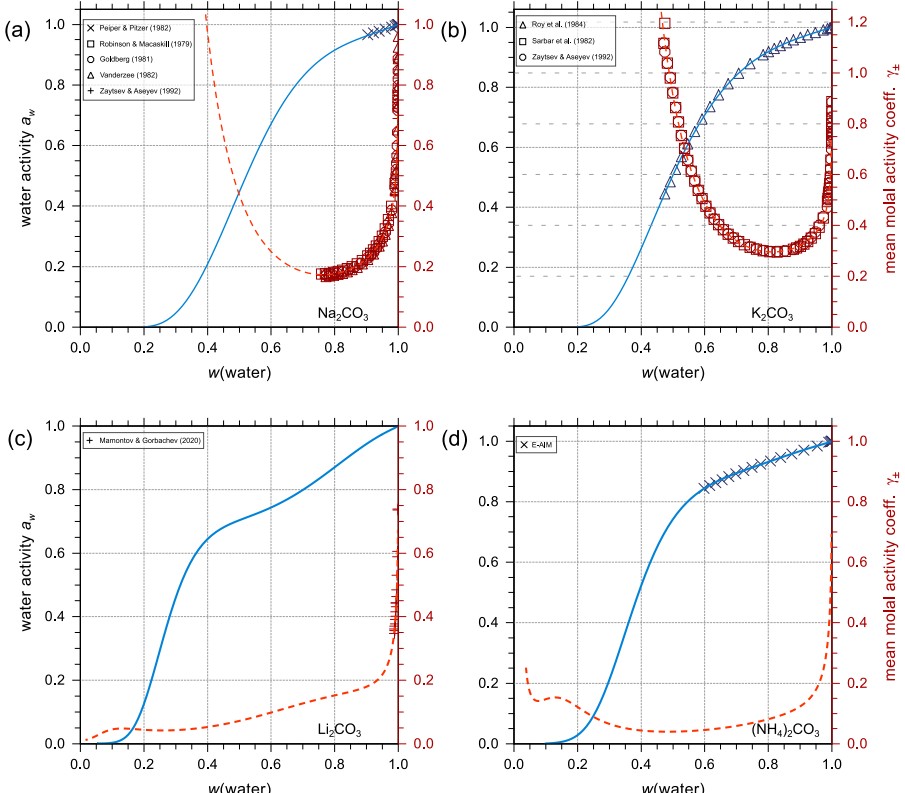

**Figure 6.** Experimental or reference model data (symbols) and AIOMFAC predictions (curves) of water activity and mean molal activity coefficients of the carbonate electrolytes **(a)** $Na_2CO_3$, **(b)** $K_2CO_3$, **(c)** $Li_2CO_3$ and **(d)** $(NH_4)_2CO_3$ in binary aqueous solutions at 298.15 K.

The agreement between AIOMFAC predictions and the experimental data from Sarbar et al. (1982a) for aqueous $NaHCO_3$ water activity covering the dilute range is excellent (see Fig. 7b). There is a slight deviation in the $HCO_3^-$ dissociation degree between the two models, which is expected as the weighting of the dissociation degree data from E-AIM is set as very small

intentionally (i.e., 0.0001). The goal of the AIOMFAC optimization for bicarbonate interactions is to make our dissociation degree comparable but not identical to that in E-AIM. This is justified because the activities involved in the associated equilibria are similar but not necessarily the same. In contrast to $H_2CO_3$ solutions, the degree of $HCO_3^-$ dissociation is generally below 0.05 in $NaHCO_3$ solutions. This makes sense since the latter solution is predicted to be of a pH value between 7.3 and 8.3, thus, a large fraction of carbonate would remain in the associated form of $HCO_3^-$. Due to the smaller concentration of $CO_{2(aq)}$

at equilibrium compared to $H_2CO_3$, the increase in the equilibrium partial pressure of $CO_{2(g)}$ for $NaHCO_3$ solutions is much more gradual than that for aqueous $H_2CO_3$.

Compared to the E-AIM online model for carbonate systems, which reports an error due to the exceedance of the supported input ion molalities, AIOMFAC is still able to solve the system of equations using the constraints on the equilibria at higher input concentrations of electrolytes. However, unlike the cases for other aqueous solutions of inorganic electrolytes for which





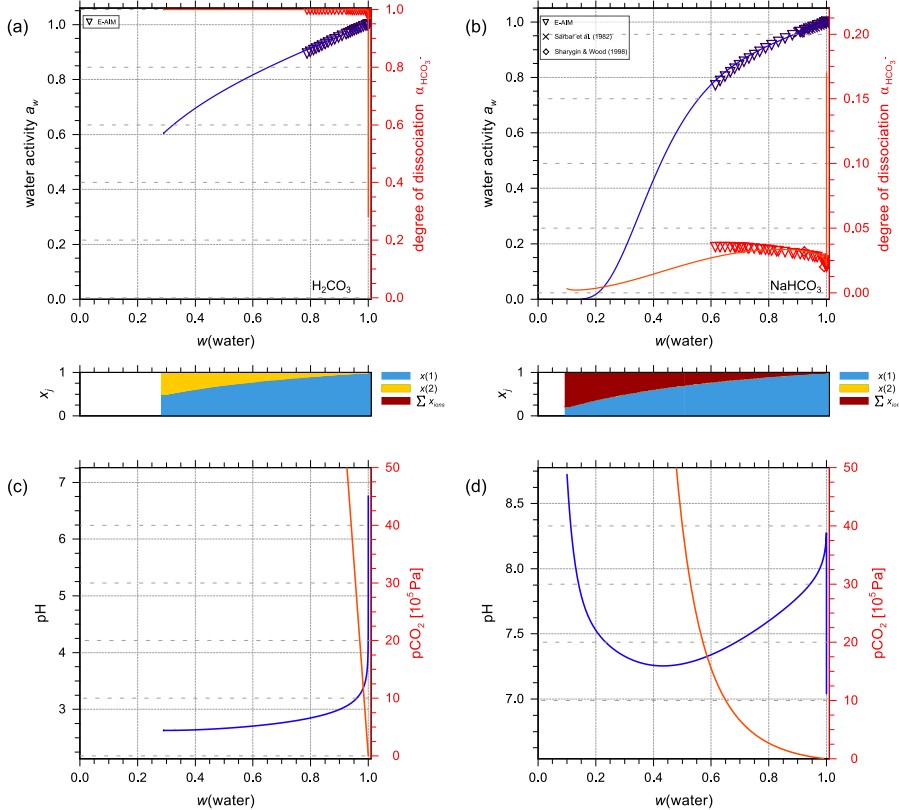

**Figure 7.** Experimental or reference (symbols) and AIOMFAC predictions (curves) of water activity and dissociation degree of $HCO_3^-$ in aqueous **(a)** $H_2CO_3$ (mixture species: water $+ CO_{2(aq)} + H^+ + HCO_3^- + CO_3^{2-} + OH^-$ ) and **(b)** $NaHCO_3$ (mixture species: water $+ CO_{2(aq)} + Na^+ + H^+ + HCO_3^- + CO_3^{2-} + OH^-$ ) solutions at 298.15 K. pH and partial pressure of $CO_{2(g)}$ in **(c)** $H_2CO_3$ and **(d)** $NaHCO_3$ (closed) systems.

AIOMFAC is able to perform well at very high ionic strengths, the validity in systems involving bicarbonate electrolytes becomes numerically limited to some degree. This is the case because the three equilibria and the mass balance equations have to be fulfilled simultaneously, it becomes numerically challenging to solve the system of equations reliably for highly concentrated conditions. Furthermore, the high partial pressure of $CO_{2(g)}$ at acidic conditions may violate the applicability of the thermodynamic equilibrium constant, which in strict terms is only valid near 1 atm total pressure. However, exceedingly high

$CO_{2(g)}$ partial pressures are only relevant for situations involving extremely high total pressures. In conclusion, AIOMFAC performs very well even in a closed system when the ionic strength becomes very large and when the associated equilibrium partial pressure of $CO_2$ exceeds atmospheric pressure by far. The highly concentrated bicarbonate solutions, under immense $CO_2$ partial pressure and hence total pressure (for which no validation data exists), correspond to the condition far from any application in Earth's atmosphere. As a result, for the conditions relevant to tropospheric aerosols, the model is considered to

perform well.





Figure 8 presents AIOMFAC predictions and reference water activity alongside dissociation degree data for aqueous $KHCO_3$, $NH_4HCO_3$, $Mg(HCO_3)_2$, and $Ca(HCO_3)_2$ solutions to showcase a selection of other bicarbonate salt systems. In the data range covered by experiments of $KHCO_3$ in water, E-AIM, the model by Meng et al. (1995) and AIOMFAC agree very well with each other. However, there are major discrepancies in water activity data generated by E-AIM and Meng et al. (1995)

beyond the experimentally accessible range. In this range (where the disagreement occurs), the AIOMFAC model parameter fit was not forced to match well with either of the model-generated reference data. Hence, the AIOMFAC curve in Fig. 8a represents a distinct interpretation of the equilibrium solution behavior at lower water contents. In other cases, such as Fig. 8b,c,d where the reference water activity data are used to fit AIOMFAC ion interaction parameters, such as $Mg^{2+}$ interacting with $HCO_3^-$ and $CO_3^{2-}$, the good agreement between AIOMFAC output and the reference $a_w$ data is as expected. Similar to the

case of aqueous $NaHCO_3$, AIOMFAC offers its own reasonable prediction of the speciation and the bicarbonate dissociation degree in the bicarbonate salt solutions.

One might argue that the closed-system scenario is unlikely to occur in the real atmosphere when water activity or relative humidity is below a certain threshold, however, the consideration of such cases is to fit the AIOMFAC interaction parameters between cations and carbonate/bicarbonate ions following the same method as E-AIM and SCAPE 2 (i.e., a closed-system

treatment).

### 4.3.2 Open-system scenario

We further performed a set of calculation scenarios for an open system, in which $CO_{2(aq)}$ is always in equilibrium with a target gas-phase $CO_{2(g)}$ mixing ratio of 400 ppm$_v$, approximately equivalent to 40.5 Pa at 1013.25 hPa total pressure. The equilibrium computation is initialized with 1 kg of water and 400 ppm$_v$ of $CO_{2(g)}$ in 1 m$^3$ of air. This setup is just

a means to solve the system, not necessarily to be representative of aerosol mass concentration. Reactions (R1) to (R4) are solved simultaneously with the equilibrium volume of the air being treated as adjustable, such that the $CO_{2(g)}$ partial pressure matches the target value. The results for aqueous $NaHCO_3$ solutions following this scenario are shown in Fig 9. For the system to reach equilibrium, the final volume of the air can be very large in contrast to the initial 1 m$^3$, because of the relatively large amount of solvent/solution used in these scenarios. Compared to a closed-system scenario (e.g., Fig. 7b), the degree of $HCO_3^-$

dissociation is much higher indicating a higher concentration of $CO_3^{2-}$ at equilibrium at the same mass fraction of water. In general, the open thermodynamic system is more basic in terms of pH than the closed one due to the shift in reaction R4 with smaller concentration of $H^+$ at equilibrium. Also, the substantially lower concentration of $CO_{2(aq)}$ present in the liquid phase affects the degree of $HCO_3^-$ dissociation strongly. The open system assumption serves the purpose of illustrating a more realistic atmospheric scenario compared to the closed system (which is essentially a system of a single aqueous phase in a

closed container with a comparably small gas phase).

### 4.3.3 Comparison with the exclusion of water dissociation

Unlike E-AIM, which makes the auto-dissociation of water a default equilibrium reaction to solve in the presence of carbonic acid, we implemented the option of including/excluding it to accommodate solving a simpler set of equations for systems/cases





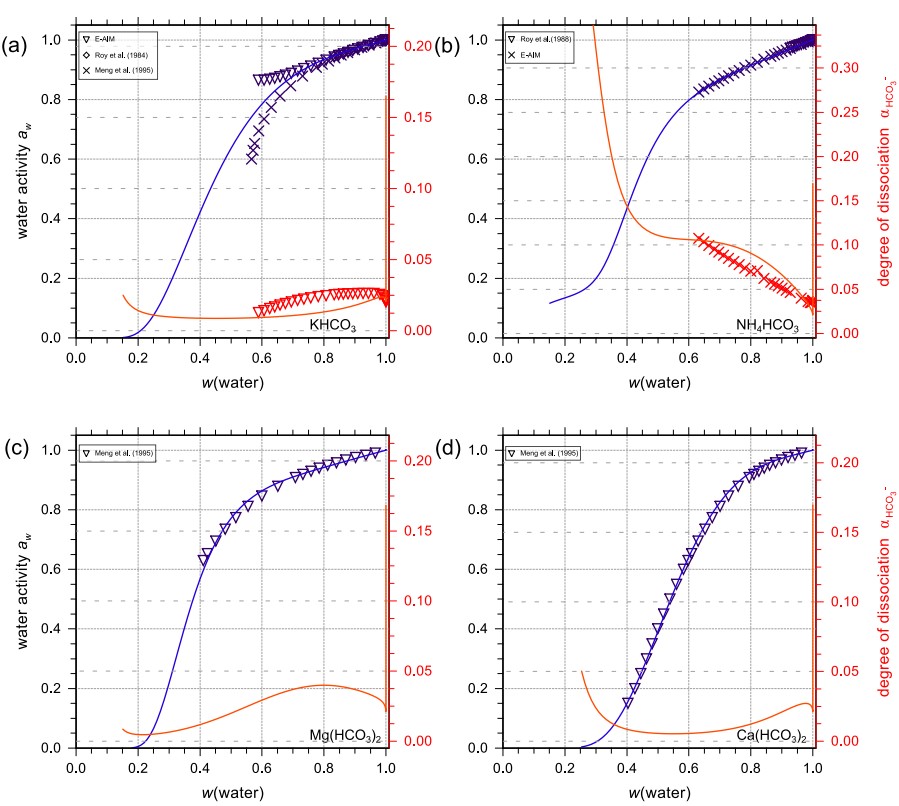

**Figure 8.** Experimental or reference (symbols) and AIOMFAC predictions (curves) of water activity and degree of dissociation of $HCO_3^-$ in aqueous **(a)** $KHCO_3$, **(b)** $NH_4HCO_3$, **(c)** $Mg(HCO_3)_2$, and **(d)** $Ca(HCO_3)_2$ solutions at 298.15 K (closed systems).

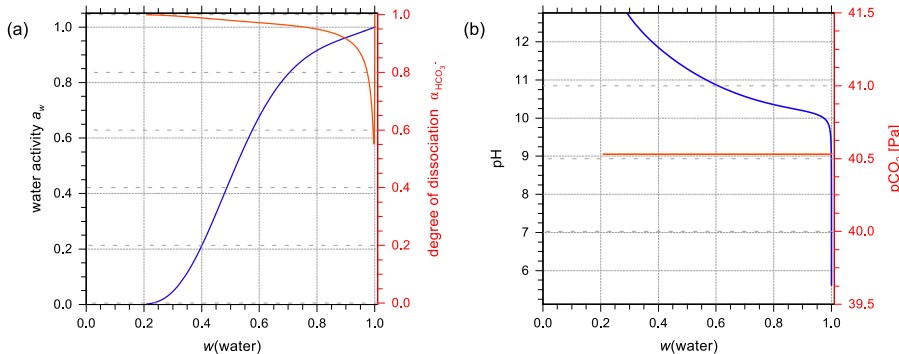

**Figure 9.** Open-system scenario with constant partial pressure of $CO_{2(g)}$ at 298.15 K. The liquid phase is an aqueous $NaHCO_3$ solution. **(a)** AIOMFAC predictions of water activity and degree of $HCO_3^-$ dissociation; **(b)** pH and partial pressure of $CO_2$.





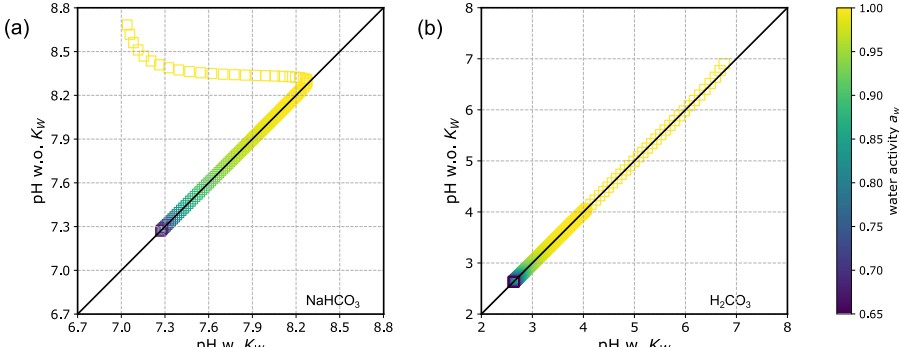

**Figure 10.** Comparison of calculated pH for the same input solution compositions but with ($x$-axis) or without ($y$-axis) the consideration of the equilibrium auto-dissociation of water ($H_2O \rightleftharpoons H^+ + OH^-$). **(a)** Aqueous $NaHCO_3$ and **(b)** aqueous $H_2CO_3$ solutions at 298.15 K. Color axis: equilibrium water activity.

in which this reaction can be neglected. When consideration of the water dissociation process is switched off, there is one

variable less ($n_{OH^-}$) and one equilibrium less (R3) to solve. In this case, the contribution of $n_{OH^-}^{max}$ to the maximum $H^+$ expression in Eqs. (15) and (18) vanishes. Figure 10 shows a direct comparison of computed pH values when the auto-dissociation of $H_2O$ is either included or excluded. The examples shown are for aqueous (a) $NaHCO_3$ and (b) $H_2CO_3$ solutions covering the water activity range from about 0.65 to 1.0. The effect of water dissociation on pH is demonstrated to be negligible at most water contents since the concentrations of both $H^+$ and $OH^-$ contributed by water dissociation are small compared

to the buffering effect on pH by other species in the system. Therefore, the simplification of omitting the auto-dissociation of water in standard AIOMFAC is justified under most conditions. An exception to this conclusion applies at very high water activity (i.e., $a_w > 0.99994$) where the pH of the solution with consideration of water dissociation tends to be closer to neutral (pH $\approx$ 7 at 298 K) due to the buffering capacity of water becoming important at extreme levels of dilution (see Fig. 10a). Hence, accounting for water auto-dissociation may have some impact on the activation of cloud condensation nuclei and/or

the pH of cloud droplets; although, the water activity values at the point of CCN activation are usually smaller than 0.9999 (typically between 0.996 and 0.9998 depending on CCN dry size (e.g., Ovadnevaite et al., 2017)). It is also evident that the effect of water dissociation becomes most relevant for solutions existing within about 1 pH unit above/below neutral pH.

### 4.3.4 Exploration of aqueous $HCO_3^- + HSO_4^-$ mixtures: open vs. closed system

In the special occasion when both sulfuric and carbonic acids are present in a solution, the incomplete dissociation of the $HSO_4^-$

ion has to be taken into account as well. An example is shown in Fig. 11a for water activity predictions from the E-AIM and AIOMFAC models for $1:1$ molar mixing of aqueous $H_2CO_3$ to $H_2SO_4$ in a closed system. Since the two models have been parameterized independently, the general agreement in terms of predicted water activity indicates that both models are capable of characterizing this multi-acid system. The dissociation degree of $HCO_3^-$ is close to 1.0 due to the solution's high acidity. Whereas the dissociation degree of $HSO_4^-$ shows a sharp decrease in the dilute concentration range and a mild fluctuation with

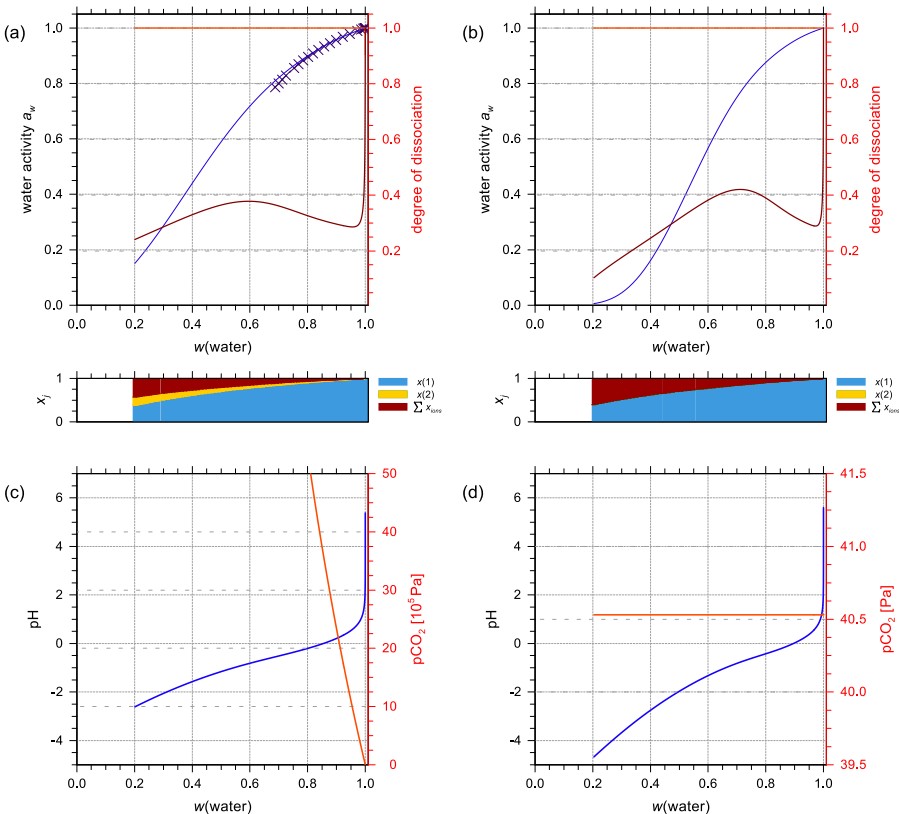

**Figure 11.** E-AIM (symbols) and AIOMFAC predictions (curves) of water activity for an aqueous solution containing $H_2CO_3 + H_2SO_4$ mixed 1:1 by moles at 298.15 K (mixture species: water $+ CO_{2(aq)} + H^+ + HSO_4^- + HCO_3^- + SO_4^{2-} + CO_3^{2-} + OH^-$). **(a)** Closed and **(b)** open system conditions. The AIOMFAC prediction of the degrees of dissociation are shown on the right y-axis for $HCO_3^-$ (light red curve) and $HSO_4^-$ (brown curve). **(c,d)** pH and partial pressure of $CO_{2(g)}$ for (c) closed system and (d) open system scenarios (400 ppm$_v$ $CO_{2(g)}$). Note the different axis scales.

increasing acid input, similar to that of a pure aqueous $H_2SO_4$ solution. This is expected since the bisulfate–sulfate equilibrium is mainly buffering the acidity of this mixture until very high dilution in water.

Analogous to the treatment discussed in Sect. 4.3.2, the solution is placed in an open system where $CO_{2(aq)}$ is set to be in equilibrium with a fixed partial pressure of $CO_{2(g)}$. Figures 11b and d show the predicted water activity, dissociation degrees, pH, and the partial pressure of $CO_2$ of the same $H_2CO_3 + H_2SO_4$ mixture in the open system case. At the same mass fraction

of water, water activity in the open system is considerably lower than that in the closed system, especially in the concentrated electrolyte range. This is largely due to the difference in speciation of the carbonate species, since in the open system the fraction of $CO_{2(aq)}$ is considerably less than that in the closed system. In addition, the solution in the open system is predicted to be much more acidic than in the closed-system case when $w(\text{water}) < 0.8$. These results demonstrate that the equilibrium composition of $HCO_3^- : HSO_4^-$ will no longer be $1:1$ in the open system and the aerosol is likely almost carbonate-free in

comparison to sulfate. Thus, under typical conditions in the atmosphere, sulfate-rich aerosols are expected to contain only very





small amounts of dissolved carbonate species. Aerosol thermodynamic models like ISORROPIA II (Fountoukis and Nenes, 2007) or MOSAIC (Zaveri et al., 2008) then often neglect carbonate species and various equilibria involved in the choice of model design due to computational efficiency and targeted applications.

## 4.4 Systems determined by alternative approaches

As discussed in Sect. 3.5, the interaction parameters for $Br^-$ and $Mg^{2+}$ with $CH_n$, $CH_n^{[OH]}$, COOH, and OH alongside calculated $B_{k,i}$ at selected ionic strengths are used as independent variables in the regression analysis to predict the unknown parameters for $I^-$ interacting with designated organic main groups. The regression resulted in an estimation of the parameters for $I^-$ with $CH_nCO$, CHO, $CH_nO$, C=C, $ACH_n$, and ACOH from the expressions

$$b_{k,I^-}^{(1)} = 0.522702 \cdot b_{k,Mg^{2+}}^{(1)} - 0.208059 \cdot b_{k,Br^-}^{(1)} + 0.01187, \tag{25}$$

$$b_{k,I^-}^{(2)} = 0.615709 \cdot b_{k,Mg^{2+}}^{(2)} - 0.145464 \cdot b_{k,Br^-}^{(2)} + 0.00374. \tag{26}$$

The correlation coefficients $R^2$ are 0.795 and 0.826, respectively, implying relatively high correlations as evaluated based on the training data. The resulting $F_{obj}$ values using fitted parameters based on experimental data and regression analysis are 3.359 and 3.690, respectively. Thus, AIOMFAC predictions of various experimental data when using the interaction parameters from the regression analysis are within 10 % error in comparison to those using parameters fitted from experimental results.

Figure 12 shows the comparison of two selected systems: the VLE of KI + water + acetone and the LLE of KI + water + ethyl acetate, when using parameters fitted from experimental data and evaluated from regression analysis. AIOMFAC predictions from the two methods show a similar level of deviation compared to the experimental data. In the case of water + acetone + KI, AIOMFAC predictions from the regression analysis actually agree better with the experimental data in the water-rich regime, therefore justifying the method as a suitable alternative in the absence of measurement data for the fit of model parameters.

Since no experimental data is available to determine the interaction parameters for iodate and main organic groups, aqueous solutions with different anions were compared to those with iodate to find the best substitution anion of iodate. Figure 13 shows a comparison between AIOMFAC calculations for binary aqueous iodate ($IO_3^-$) and nitrate ($NO_3^-$) solutions, where the two electrolytes have the cation in common (different panels for the cations $Na^+$, $K^+$, $Li^+$ and $H^+$). Due to the similarities in ion size and thermodynamic behavior between the two anions, the $IO_3^- \leftrightarrow$ organic main group interaction parameters can be

estimated as a simple approximation, by

$$IO_3^- \leftrightarrow \text{ organic main group } \approx NO_3^- \leftrightarrow \text{ organic main group.} \tag{27}$$

For the same reason, aqueous $NH_4IO_3$ and $Ca(IO_3)_2$ solutions, which cannot be added based on experimental data, adopted this substitution as well (e.g., $Ca^{2+} \leftrightarrow IO_3^- \approx Ca^{2+} \leftrightarrow NO_3^-$). We refer to such parameter substitutions based on similarity in behavior as the analogy approach. Although simpler and likely less reliable than the regression-based approach, the analogy



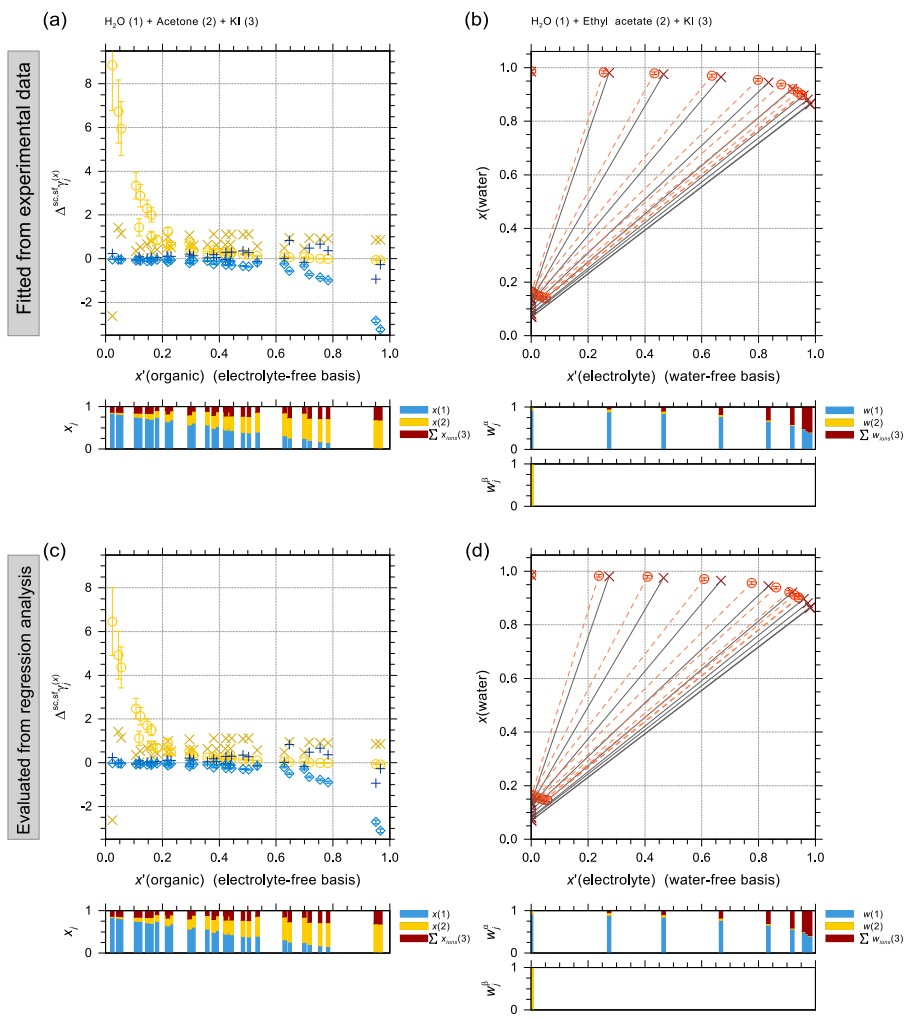

**Figure 12.** Comparison of experimental data ($\times$, $+$) and predicted ($\bigcirc$, $\diamond$) values for ternary systems using two different approaches for the iodide interaction parameters within AIOMFAC. Predictions from the left and right panels are based on fitted parameters from experiments (top) and our regression approach (bottom). **(a, c)** VLE of water (1) + acetone (2) + KI (3) at 322–360 K; experiments by Al-Sahhaf and Jabbar (1993). **(b, d)** LLE of water (1) + ethyl acetate (2) + KI (3) at 298 K; experiments by Al-Sahhaf et al. (1999). The composition bars show the mass fractions (b,d) or the mole fractions (a,c) of the components with respect to dissociated salts.





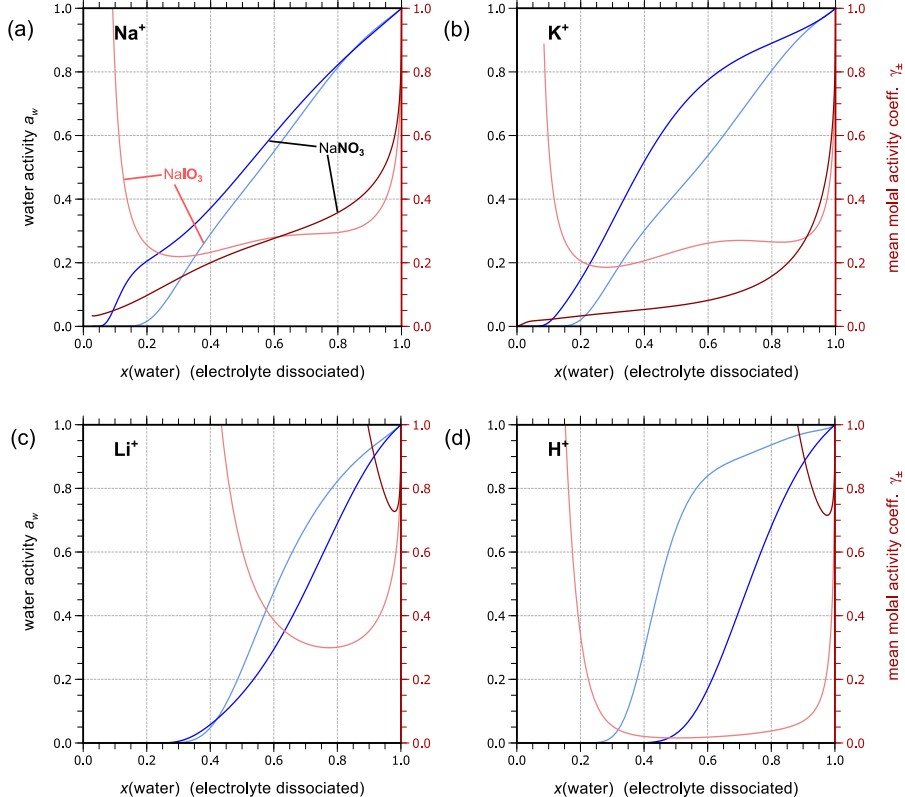

**Figure 13.** Comparison of AIOMFAC predictions of water activity and mean molal activity coefficients at 298.15 K for different binary aqueous solutions of either iodate or nitrate electrolytes with a common cation. Light-colored curves show the iodate case while darker shades denote the nitrate solutions. **(a)** $NaIO_3$, $NaNO_3$, **(b)** $KIO_3$, $KNO_3$, **(c)** $LiIO_3$, $LiNO_3$, **(d)** $HIO_3$, $HNO_3$.

approach is the best option when there are not sufficient sets of experimental data available to determine the linear regression coefficients.

Based on the analogy approach, the parameters for organic main groups interacting with $CO_3^{2-}$ are substituted by those of the same organic main groups interacting with $SO_4^{2-}$ due to the lack of experimental data. Figure 14 shows the estimated LLE and SLE calculations for carbonate salts + water + organic compounds using substituted parameters for $CO_3^{2-} \leftrightarrow$ organic 645 main group interactions alongside the experimental data. Panel (a) shows the LLE of water + 1-propanol + $Na_2CO_3$ at 315.15 K and panel (b) presents the model–measurement comparison of phase composition for a LLE system of water + 1-butanol + $K_2CO_3$ at 298.15 K. AIOMFAC performs very well in these cases considering the large sensitivity to ion/salting-out effects in predicting liquid–liquid equilibria. Figure 14c shows the model prediction of $Na_2CO_3$ mole fraction at the salt solubility limit in this ternary mixture of water + acetone + $Na_2CO_3$ at 295.15 K. The model prediction agrees quite well 650 with the measurement, especially with the sharp decrease in the salt solubility at lower water content. Panel (d) of Fig. 14 shows another example of a LLE phase composition diagram, in this case the LLE of water + ethanol + $K_2CO_3$ at 298.15 K. The experiments indicate the existence of liquid–liquid phase separation in this system, as shown by the tie-lines connecting





coexisting phases, but the model predicts a single liquid phase to remain stable for the same overall mixture compositions. This may be attributed to the use of ion ↔ main group interaction parameters from sulfate here, since similar predictions were made for LLE cases of sulfate salts interacting with ethanol (Zuend et al., 2011). The mostly satisfying results achieved for the calculation of aqueous carbonate or iodide salts mixed with organics, justify the use of such analogy-based parameter substitutions when other means for determining the interaction parameters are not possible. Such cases are sometimes found even if a few measurement datasets are available, as is, e.g., the case for the carbonate salt systems shown in Fig. 14. Given the lack of a sufficiently diverse database in terms of organic main groups and their relative abundances in molecules of measurement data, constraining the model parameters based on only a few datasets will often lead to a substantially biased set of parameters that would likely result in weak predictability for systems not covered by the training data. Hence, the necessity for using alternative estimation methods. However, the parameters determined from the alternative methods may be subject to revision once more experimental data pertaining to the ions/organic groups in question become available in future.

## 5 Atmospheric implications

The hygroscopic growth behavior and cloud droplet activation ability of aerosol particles are two (among many) important composition-dependent properties of ambient particles. These properties are directly dependent on the mixing and gas/liquid/ solid phase equilibrium thermodynamics of aerosols (e.g., Seinfeld and Pandis, 1998; Pankow, 2003; Hersey et al., 2013). Hence, there is a need for (predictive) thermodynamic aerosol models to serve as key components in box models and chemical transport models for the simulation of atmospheric chemistry and physics (e.g., Zaveri et al., 2008; Pye et al., 2020). Importantly, ambient aerosols are typically multicomponent organic–inorganic mixtures containing several types of inorganic ions and hundreds to millions of different organic compounds (Goldstein and Galbally, 2007; Kroll et al., 2011). Such complex systems may exhibit phase separation or otherwise non-ideal mixing, critically affecting the gas–particle partitioning of semivolatile components, including water (e.g., Zuend and Seinfeld, 2012; Ovadnevaite et al., 2017; Pye et al., 2018). By extending the AIOMFAC model with new species and revised interaction parameters, a wider variety of atmospherically relevant species can be covered enabling more detailed multi-phase box model calculations. In turn, improvements in predicted aerosol and cloud droplet acidity, aerosol mass concentrations and phase behavior serve the interpretation of laboratory and field measurements (Pye et al., 2018, 2020) as well as the development and assessment of reduced-complexity aerosol treatments in air quality and climate models (Rastak et al., 2017).

### 5.1 Iodine and carbonate species in aerosols or cloud droplets

With the newly implemented parameters describing the interactions between aqueous iodine or carbonate electrolytes and organic compounds, the AIOMFAC model is capable of predicting the mixing behavior within aerosol phases containing these species (among many others). Furthermore, while the uncertainty of parameters determined using the substitution method might be relatively large, such method adoption allows for the completion of AIOMFAC's interaction parameter matrix (Fig. 1),



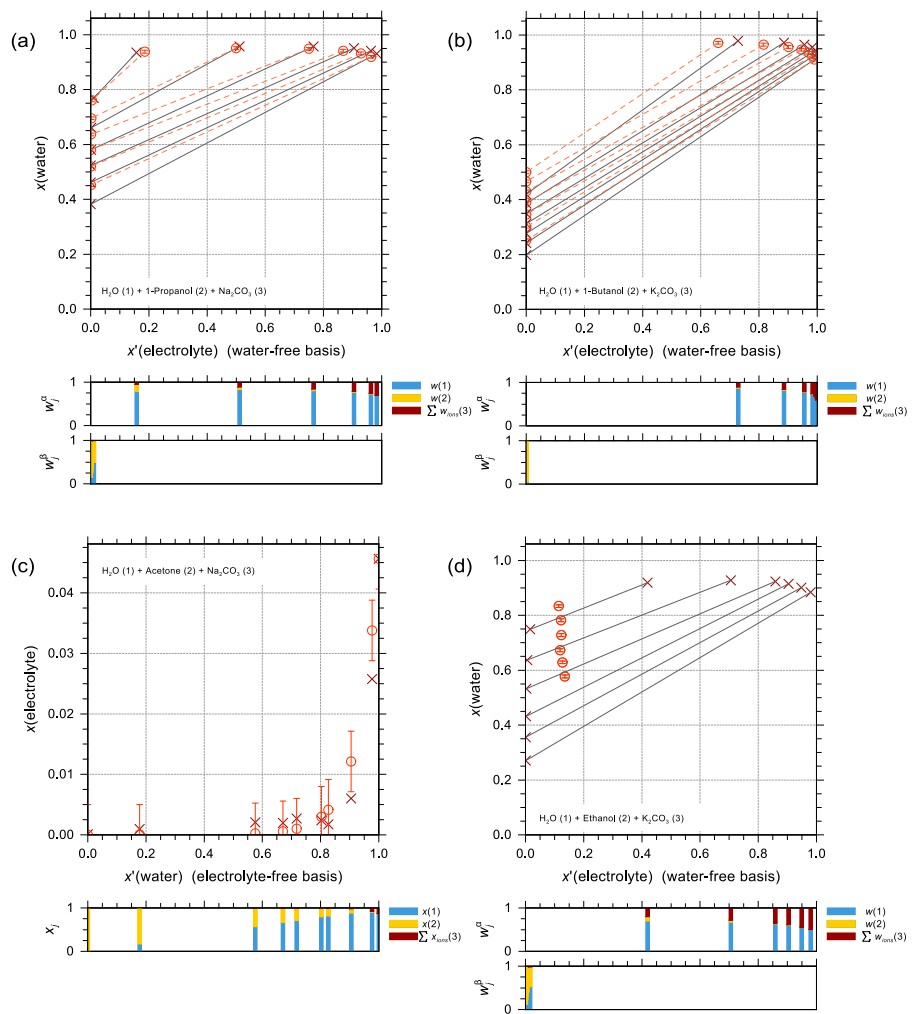

**Figure 14.** Experimental ($\times$) and AIOMFAC-predicted ($\bigcirc$) equilibrium phase compositions of water + organic compound + carbonate salt systems with model sensitivity shown as error bars. **(a)** LLE of water (1) + 1-propanol (2) + $Na_2CO_3$ (3) at 313 K; experiments by Gerlach and Smirnova (2016). **(b)** LLE of water (1) + 1-butanol (2) + $K_2CO_3$ (3) at 298 K; experiments by Fu et al. (2020). **(c)** SLE of $Na_2CO_3$ in a water (1) + acetone (2) + $Na_2CO_3$ (3) system at 295 K; experiments by Ellingboe and Runnels (1966). **(d)** LLE of water (1) + ethanol (2) + $K_2CO_3$ (3) at 298 K; experiments by Salabat and Hashemi (2007). The AIOMFAC-LLE model predicts no phase separation for this system (d). The composition bars show the mass fractions (or mole fractions in c) of the components with respect to dissociated salts.





improving the model range of applicability. Moreover, for iodine and carbonate systems, specific atmospheric physical or
chemical processes can be explored using the extended AIOMFAC modeling framework.

Recent measurements have shown different soluble iodine speciation in fine mode and coarse mode aerosol particles (Baker
and Yodle, 2021). Generally, higher proportions of iodide and soluble organic iodine (SOI) are associated with fine mode
aerosols, while coarse mode aerosols are dominated by iodate instead. In terms of acidity influence, alkaline sea spray or
mineral dust aerosols contain higher ratios of iodate, while acidic particles are rich in iodide and SOI. However, due to the lack
of direct pH measurements of the sample aerosols, the new findings are not sufficient to explain the possible links between
acidity and speciation results. With the addition of iodine system into AIOMFAC, detailed iodine speciation modeling can be
done by coupling AIOMFAC with a kinetic chemistry and mass transfer model. This, together with field measurements on gas
and aerosol phase composition and pH, can have the potential of explaining atmospheric iodine speciation.

We also show that if an aerosol particle is composed of both bicarbonate and bisulfate, at equilibrium, the effects of sulfate
are mostly dominating over carbonate as shown in Fig. 11. However, in a cloud droplet, when the relative humidity is close to
1.0, it is important to consider the effects of carbonic acid or relevant carbonate/bicarbonate salts. The addition of the carbonate
system and the associated equilibria to AIOMFAC can serve as a more comprehensive thermodynamic model to couple with
aerosol and cloud-water chemistry models for better activity and reaction rate simulation quality.

## 5.2 Computations of CCN activation properties

A key atmospheric impact of fine and ultrafine mode aerosol particles stems from their ability to act as cloud condensation nu-
clei. The number–size distribution and hygroscopic properties of sufficiently large aerosol particles impact cloud droplet num-
ber concentrations during cloud formation, ensuing cloud microphysics and aerosol–cloud–radiation effects (e.g., Lohmann
and Feichter, 2005; Carslaw et al., 2013; Gettelman, 2015). Clouds in marine or other pristine environments characterized by
relatively low aerosol mass and number concentrations are typically more sensitive to the hygroscopicity and CCN activation
properties of the present particles (e.g., Carslaw et al., 2013; Fossum et al., 2018, 2020). Iodide and iodate ions are expected to
be more prevalent in marine boundary layer aerosols (on $pmol\,m^{-3}$ levels), although their concentrations and speciations are
only infrequently measured in the field (Baker, 2004; Baker and Yodle, 2021).

Based on the introduced AIOMFAC model extension, thermodynamic CCN activation calculations for systems containing
iodine or carbonate species are now possible. This includes computations of Köhler curves and associated CCN activation
properties for sodium iodide or sodium carbonate particles mixed/coated with suberic acid at 293 K to compare with corre-
sponding measurement reported by Davies et al. (2019). Such computations involve predictions of particle size at different
RH levels, which requires knowledge or estimations of apparent molar volumes or densities of the corresponding aqueous
solutions. The density of pure crystalline NaI is $3670\,kg\,m^{-3}$ (Yaws, 2009). Since it is not possible to measure the density of
pure liquid NaI, an apparent liquid-state density of $4230\,kg\,m^{-3}$ is used, which is derived from the apparent molar volume
extrapolation by Zhuo et al. (2008). The solid-state density of pure $Na_2CO_3$ is $2540\,kg\,m^{-3}$ (Yaws, 2009). The pure liquid-
state density of $Na_2CO_3$ is assumed to be of the same value, since liquid-state data is unavailable. The rest of the relevant
physical parameters used in the AIOMFAC calculation are the same as those listed in Table 1 of Davies et al. (2019). Similar





to the AIOMFAC-based predictions for CCN activation properties carried out in the study by Davies et al. (2019), three differ-ent modeling variants are adopted here. The full gas–particle and liquid–liquid equilibrium calculation (AIOMFAC-EQUIL)

allows all components to partition between the gas phase and up to two liquid phases to determine the equilibrium state. In con-trast, the variants described as AIOMFAC-CLLPS assume that the inorganic ions and the organic components always reside in different liquid phases. AIOMFAC-CLLPS is further split into two variants, both assuming a core–shell particle morphology: one is labeled "with org. film" in which water is not allowed in the organic-rich shell for the purpose of hygroscopic growth and surface tension computations. This leaves the surface of the particle as an organic film phase, strongly affecting the particles'

surface tension when full surface coverage is possible. The other variant is denoted as AIOMFAC-CLLPS (without org. film), which allows water to partition into the organic-rich shell phase (in addition to the aqueous inorganic core phase); see Davies et al. (2019) for further details.

AIOMFAC predictions of the critical supersaturation for CCN activation, $SS_{crit}$, under different conditions in aqueous particle systems containing suberic acid and NaI or $Na_2CO_3$ as the particle core are listed in Tables S1 and S2 in the SI. In the case

of NaI-containing particles, Fig. 15a shows that the different model predictions of $SS_{crit}$ are systematically higher than those determined from the experiments using particles of 50 nm dry diameter ($D_{dry}$) across different volume fractions of suberic acid. Among the different model variants, it is evident that the $SS_{crit}$ predictions based on AIOMFAC-CLLPS (with org. film) lead to lower $SS_{crit}$ values with increasing organic dry volume fractions ($f_{org}$) compared to the other two model variants, a finding consistent with those for similar particles containing ammonium sulfate as inorganic salt (Davies et al., 2019). However,

none of the AIOMFAC variants are able to capture the measured data because of the systematic offset already present for the organic-free case of NaI particles. Considering that our model is very well parameterized for the modeling of the binary aqueous NaI solution, potential errors in the aqueous electrolyte + organic mixture predictions are clearly unable to explain the observed model–measurement deviation at low $f_{org}$. One possible reason to account for the shown difference is the uncertainty associated with the apparent liquid density of NaI used in the model. For a 15 % reduction in the pure liquid-state density

of NaI to 3595.5 $kg\,m^{-3}$, the resulting $SS_{crit}$ value of the binary water + NaI system for $D_{dry} = 50$ nm is 0.495 %, which remains substantially higher than the measured $SS_{crit} = 0.38$ % value. After a series of exploratory calculations, results from AIOMFAC-EQUIL and AIOMFAC-CLLPS (without org. film) seem to agree better with the experiments when assuming a dry diameter of $D_{dry} = 59$ nm, as shown in Fig. 15b. Taking the nonspherical shape of dry NaI particles into consideration, it is unlikely realistic for the shape correction factor to be as large as 10 %. At present, we are unable to satisfactorily explain

the observed differences for the organic-free system – and other similar systems studied by Davies et al. (2019) did not exhibit such an offset. In addition to this conundrum, none of our AIOMFAC-based model variants are able to simulate the zigzag behavior of the measured $SS_{crit}$ data near $f_{org} = 0.5$, a robust feature in this system, as confirmed by repeated measurements. Capturing such an intriguing $SS_{crit}$ curve phenomenon may require a more sophisticated treatment of composition-dependent particle morphology and related surface tension.

As stated previously, the parameters for $CO_3^{2-} \leftrightarrow$ organic main group interactions are adopted via substitution from those for $SO_4^{2-} \leftrightarrow$ organic main group interactions. In the case of the binary aqueous $Na_2CO_3$ system, the AIOMFAC prediction of the critical supersaturation for $D_{dry} = 50$ nm is 0.330 %, which is in excellent agreement with the reported $SS_{crit} = 0.33$ %



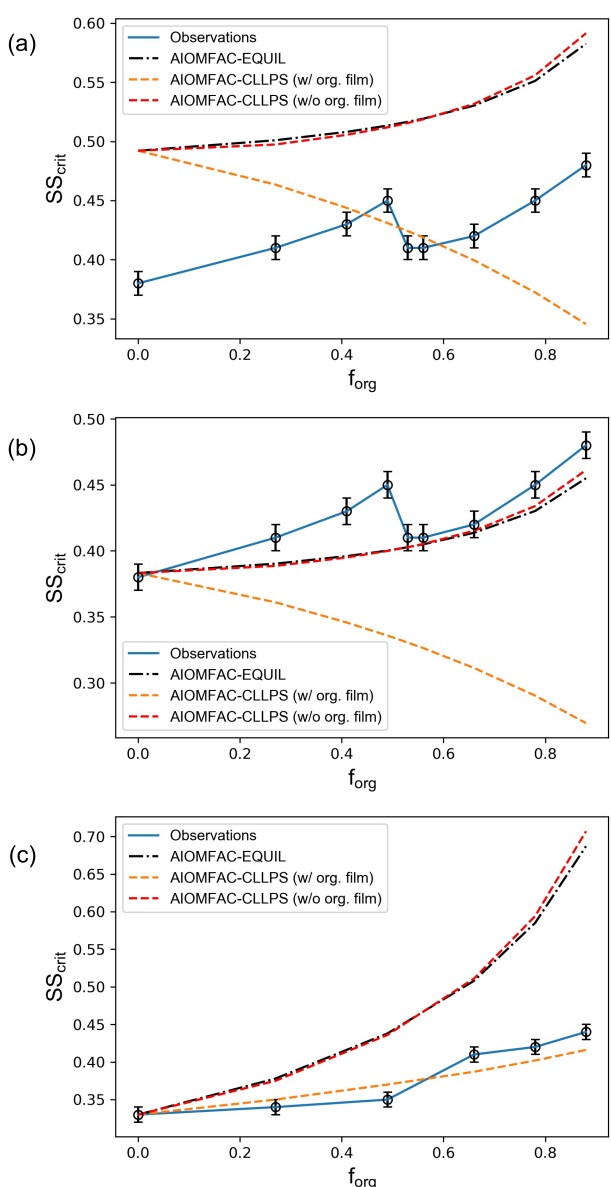

**Figure 15.** Comparison of critical supersaturation ($SS_{crit}$) data between measurements and predictions from different AIOMFAC-based model variants. **(a)** Measured data ($\bigcirc$, blue line) and AIOMFAC predictions (curves) of $SS_{crit}$ for cloud droplet activation of NaI particles coated/mixed with suberic acid for $D_{dry} = 50$ nm containing different volume fractions of suberic acid ($f_{org}$). **(b)** Same system as in (a) but using an adjusted particle dry diameter of $D_{dry} = 59$ nm in case of the model predictions. **(c)** Similar to (a) but with $Na_2CO_3$ as inorganic component for $D_{dry} = 50$ nm. Experimental data by Davies et al. (2019).





value from the experiment. Among the three different model variants, the AIOMFAC-CLLPS (with org. film) prediction agrees best with the measurements, as shown in Fig. 15c. The same conclusion was also made for a similar system of suberic acid coated $(NH_4)_2SO_4$ particles in the work by Davies et al. (2019). At first glance, this result seems to be the least intuitive, as one may expect that some amount of water would likely partition into the organic film phase. At present, none of the model variants accounts fully for the process of bulk–surface partitioning in the absence of liquid–liquid phase separation at high relative humidity (which is the case in these suberic acid containing systems). Accounting for composition and size-dependent bulk–surface partitioning alongside phase separation may aid in explaining the observed behavior in future work.

Notwithstanding the remaining challenges in modeling CCN activation properties of rather peculiar systems (such as those of Fig. 15), the coverage of several carbonate and iodide species by AIOMFAC has further extended the thermodynamic modeling capabilities of relevant mixed organic–inorganic aerosol systems. It provides the community with a predictive tool for the mixing thermodynamics and hygroscopicity of relevant laboratory systems as well as more complex samples of iodine- and carbonate-containing tropospheric aerosols.

## 6 Conclusions

The latest extension of the AIOMFAC model, developed and introduced in this work, adds the ions $I^-$, $IO_3^-$, $CO_3^{2-}$, $HCO_3^-$, $OH^-$, and $CO_{2(aq)}$ in mixtures containing water and a variety of organic compounds. The thermodynamic conditions necessary to be considered for systems containing the added carbonate species involve coupled equilibria and material constraints. We discussed how solving numerically the associated system of algebraic equations can be challenging under highly acidic or highly basic pH conditions, under which the molalities of certain ions, such as $OH^-$ or $H^+$ become negligibly tiny. Approaches to navigate those conditions were implemented, enabling the use of AIOMFAC for computations of carbonate/bicarbonate/$CO_2$ equilibria in open- or closed-system scenarios, applicable to a wide range of concentrations relevant to atmospheric aerosols, cloud droplets and/or other aqueous environments.

During the simultaneous fitting of AIOMFAC cation–anion and ion–organic main group interaction parameters, numerous experimental datasets were accessed and compared with the model predictions. The agreement between AIOMFAC calculations and the majority of experiments ranges from satisfactory to excellent. Relatively large deviation exists in some cases, for which we discussed potential reasons related to the use of a group–contribution method. Due to the lack of experimental data covering certain aqueous cation–iodate systems, as well as the interactions of the newly introduced ions with several organic functional groups, alternative approaches for parameter estimation were explored. Among those approaches are the use of linear regression to determine iodide ion interactions with certain organic functional groups based on the interaction coefficients of a set of other ions with those groups. Adoption of such methods in practice means that we were able to provide estimates for model parameters describing most of the possible binary interactions involving the new species. This is considered very useful to close otherwise existing gaps in the model's interaction matrix (Fig. 1). It enables the AIOMFAC model to provide predictions for a wide range of multicomponent, multifunctional systems of interest in atmospheric chemistry – even when suboptimal in terms of the underlying experimental database. In addition to the adoption of such estimation methods,


we also conducted new bulk water activity and EDB measurements to cover systems involving iodide or iodate electrolytes in greater detail. Nevertheless, the current parameterization of AIOMFAC for certain carbonate and iodine species is limited and uncertain to some extent, offering the potential for improvements in future work – subject to the availability of suitable new thermodynamic equilibrium data. In total, 285 new interaction parameters were determined, describing interactions for 35 cation–anion and 55 ion–organic main group pairs.

As an example of an application of AIOMFAC using the newly determined interaction parameters, we computed the critical supersaturation as a function of dry particle size and organic fraction for suberic acid coated salt particles. The AIOMFAC predictions of $SS_{crit}$ for $Na_2CO_3$ particles were found to be in good agreement with available measurements, while similar predictions for NaI particles showed a systematic offset compared to the measurements. Future work will focus on incorporating partial dissociation of various organic acid species in AIOMFAC.

*Code and data availability.* The new model extension will be included in an upcoming version of AIOMFAC-web, which can be run at https://aiomfac.lab.mcgill.ca.

*Author contributions.* AZ conceptualized the project. HY and AZ developed the methodology, evaluated the data, and wrote the software. HY carried out a part of the water activity measurements at McGill University with the assistance of AB, BJW, and TCP. JD, LK, and UKK performed the EBD and water activity measurements at ETH Zurich. HY analyzed the data and created the visualizations. HY wrote the manuscript with contributions from all co-authors.

*Competing interests.* The authors declare that they have no conflict of interest.

*Financial support.* This research has been supported by the Fonds de recherche du Québec – Nature et technologies (FRQNT), grant no. PR-286433, and the Natural Sciences and Engineering Research Council of Canada (NSERC), grants no. RGPIN-2014-04315 and RGPIN-2021-02688. LK acknowledges funding support by the Swiss National Science Foundation (SNSF), grant no. CRSII5-189939.





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
