# Peer review of "Extension of the AIOMFAC model by iodine and carbonate species: applications for aerosol acidity and cloud droplet activation"

_Atmospheric Chemistry and Physics, 2021_

## Author Comment (AC1)

**Authors' response to referee #1**

**Hang Yin et al.**

**November 2021**

We would like to thank referee #1 for the effort in providing this review. In the following, we repeat each comment by the referee (in black font) followed by our response and, where applicable, related changes to the revised version of the manuscript in blue font. Page and line numbers stated are those from the original manuscript.

General comments:

This manuscript describes the further development of the AIOMFAC model to include additional chemical species, such as carbonates and iodine. Using literature thermodynamic data and new experimental observations, the authors derive estimates for the interaction parameters between ionic species and certain organic functional groups. The authors report excellent agreement between model and experimental results for inorganic mixtures, and good agreement with organic mixtures at high humidity. Overall, this work provides the community with a detailed description of the model development and will allow for a broader application of the AIOMFAC model to understanding the thermodynamics of chemically complex systems. It is appropriate for this journal, well-written and can be published with minimal changes, as detailed below.

Specific comments:

1. On line 279, the authors describe the assumption that $CaSO_4$ will form a solid. This is reasonable, as the authors point out, for most RH conditions, but likely not cloud forming conditions, where even low solubility species will likely enter solution. How does the assumption of solid formation impact AIOMFAC predictions for CCN activation estimates? What other species are assumed to form solid phases only?

Authors' response: At present, $CaSO_4$ is the only compound considered to form a solid phase as an *a priori* assumption (and thereby removing some or nearly all of the sulfate from an aqueous solution). However, this treatment can be modified for calculations under high dilution, as those mentioned by this reviewer. In a single-phase calculation scenario, since RH or water activity is an output of an AIOMFAC calculation, it is typically simpler to assume complete precipitation of $CaSO_4$ regardless of the water content or equilibrium RH. However, for gas–particle partitioning or CCN activation calculations, typically carried out with a targeted RH as an input, a RH threshold (e.g., 98 %) can be set within our AIOMFAC-based equilibrium model, such that at RH levels above this threshold $CaSO_4$ is allowed/assumed to fully dissolve into the aqueous solution. Pye et al. (2020), in their section 4.2, provide a related discussion concerning the treatment of $CaSO_4$ and other potential

solids within several thermodynamic aerosol models.

Changes to manuscript: On line 281 we add the following: "For systems involving $Ca^{2+}$ and $SO_4^{2-}$ ions at high RH in gas–particle equilibrium computations, the treatment of $CaSO_4$ can be modified by introducing a RH threshold, e.g. 98 %, above which the $CaSO_4$ is assumed to be fully dissolved in an aqueous phase. Such an approach was adopted for AIOMFAC-based equilibrium calculations in the study by Pye et al. (2020)."

2. Suggest Figure 4 change the x-scale range to better show the relevant data – currently the data points span only $1/5^{th}$ or less of the width of the figures, making its hard to judge what the data shows.

   Authors' response: In addition to the experimental data/model prediction comparison, we wanted to highlight the effect of salt addition compared to the salt-free case over the whole concentration range. A zoomed-in version which only shows the experimental range have been added to the supplemental information.

   Changes to manuscript: Figure S4 has been added to the supplemental information document.

3. Figure 5A needs further explanation in the text and/or figure caption – what is Also, these data are reported at 351 – 366 K – this is very close to the boiling point of water-ethanol, depending on the mixture composition. It's unclear how to interpret these data and a greater description is necessary.

   Authors' response: This is an isobaric vapor–liquid equilibrium data set for which measurements were conducted at 101.3 kPa, i.e. approximately 1 atmosphere pressure generated by the combination of water and ethanol vapors (in a closed system), for different mixture compositions. This is why the temperature is relatively high and varying over small range. Detailed description of this type of data set and associated treatment for model parameter optimization has been discussed in Zuend et al. (2011). A summary of how such data sets were processed is given in Section 3.4, including Eq. (22) of our manuscript. We modified the figure caption to reflect the information more clearly in the revised version.

   Changes to manuscript: Caption of Fig. 5: "(a) Isobaric VLE data of water (1) + ethanol (2) + KI (3) at 351–366 K; experiments by Chen and Zhang (2003)."

4. Line 744 – "unlikely realistic" is awkward and confusing phrasing. Please amend.

   Authors' response: We agree, this phrase was confusing. Meanwhile, we have found a mistake in the baseline calculation of the critical supersaturation values used for the water + suberic acid + NaI system. The mistake only affected the post-processing of calculations for this specific system. The corrected values for the organic-free water + NaI agree well with the experimental data for a dry diameter of 50 nm using the listed densities and without the need of any dynamic shape correction factors. As such, Fig. 15 and the whole related text section have been updated.

   Changes to manuscript: Lines 731–746 regarding the model prediction/measurement offset

in the water + suberic acid + NaI case will be amended. Figure 15 (panels a,b) will be updated. A related sentence on line 794 of the Conclusions section has been rephrased.

Technical Points:

1. Labels on graphs are generally on the small side and would likely be illegible on printed page.

   Authors' response: All the figures are high-quality vector graphics which retain clarity even with extensive zooming-in. We have enlarged several of the figure labels. Furthermore, the positioning and scaling of the figures in the final publication will likely aid in increasing the size of the labels.

   Changes to manuscript: Labels in Fig 3, 6 , 7, 8 have been enlarged.

**References**

Chen, W.-M. and Zhang, Y.-M.: Vapor–Liquid Equilibria for Alcohol–Water–KI/NaAc Systems, J. Chem. Eng. Chin. Univ., 17, 123–127, https://doi.org/10.3321/j.issn:1003-9015.2003.02.002, 2003.

Pye, H. O. T., Nenes, A., Alexander, B., Ault, A. P., Barth, M. C., Clegg, S. L., Collett Jr, J. L., Fahey, K. M., Hennigan, C. J., Herrmann, H., Kanakidou, M., Kelly, J. T., Ku, I. T., McNeill, V. F., Riemer, N., Schaefer, T., Shi, G., Tilgner, A., Walker, J. T., Wang, T., Weber, R., Xing, J., Zaveri, R. A., and Zuend, A.: The acidity of atmospheric particles and clouds, Atmos. Chem. Phys., 20, 4809–4888, https://doi.org/10.5194/acp-20-4809-2020, 2020.

Zuend, A., Marcolli, C., Booth, A. M., Lienhard, D. M., Soonsin, V., Krieger, U. K., Topping, D. O., McFiggans, G., Peter, T., and Seinfeld, J. H.: New and extended parameterization of the thermodynamic model AIOMFAC: calculation of activity coefficients for organic-inorganic mixtures containing carboxyl, hydroxyl, carbonyl, ether, ester, alkenyl, alkyl, and aromatic functional groups, Atmos. Chem. Phys., 11, 9155–9206, https://doi.org/10.5194/acp-11-9155-2011, 2011.

---

## Author Comment (AC2)

**Authors' response to referee #2**

Hang Yin et al.

November 2021

We would like to thank referee #2 for the effort in providing this review. In the following, we repeat each comment by the referee (in black font) followed by our response and, where applicable, related changes to the revised version of the manuscript in blue font. Page and line numbers stated are those from the original manuscript.

General comments:

The manuscript by Yin et al. describes the extension of the AIOMFAC model to iodine and carbonate species. These chemical species are important for sea spray and mineral dust particles in the atmosphere. The extension will play a critical role in future thermodynamic studies on coarse mode particles. Generally, the scientific approach of the manuscript sounds. The output of the developed model agrees well with experimental data when it is available. As the authors mention in the manuscript, experimental data for the chemical species are scarce. I hope that the manuscript will stimulate experimental scientists in the area to conduct high-quality study on the related chemical species, and the authors will update the model parameters in the future when it is needed. The manuscript is well written, and the topic is within the interest of the readers of the journal. I suggest publication of this manuscript after addressing the comments

Specific comments:

1. Table 2: It would be more informative if the authors could add further information about the experimental details of the references (e.g., temperature, range of ionic strength, and experimental method).

   Authors' response: Two extra columns regarding the experiment temperature and ionic strength range have been added to Tables 2 and 6. Conceptual descriptions of various experimental methods are discussed in Section 3.4 and a more detailed description of our own measurements is provided in the supplemental information.

   Changes to manuscript: Two columns $T$ (K) and $I$ $(\mathrm{mol\,kg^{-1}})$ have been added to Tables 2 and 6.

2. I understand that determination of weighting factors for this type of study needs to be arbitrary. It would be ideal to have a little bit more detailed descriptions on how the weighting factors have been decided.

   Authors' response: Additional discussion has been added in the manuscript to address the

rationale behind the (initial) weighting factor assignment; see the following changes.

Changes to manuscript: Line 187: "The initial weightings of the datasets for the fitting process were set to unity unless otherwise specified (see footnote in Table 2)."
Line 209: "The initial weightings of individual datasets were first determined based on an estimated relative uncertainty associated with the experimental method. For example, water activity measurements from bulk solutions were generally assigned with weighting value units higher than those from EDB measurements. After a first round of the fitting process, the relative contributions from different datasets to the total objective fit function value were evaluated. Datasets which indicated potential inconsistencies with other datasets for a certain system, as assessed from their large (or contradicting) contributions to the objective function, were carefully checked for potential errors in input files and associated data. For datasets considered valid but of high objective function contribution, we lowered the initial weighting such that their contribution became more similar to the median objective function value contribution of different datasets. Such manual, iterative inspections of fit progress, associated graphical data comparison, and dataset weighting adjustments, aid in avoiding potential issues of large biases in parameter estimation, which may occur when the fit error is dominated by only a small subset of datasets."

3. L372: 'Instead, we estimate the interaction parameters for carbonate ions and organic compounds based on those for sulfate ions due to the similar ion size and electric charge.' The statement might make sense for the Coulombic force. I wonder if the assumption is valid for middle- and short-range forces.

Authors' response: The short-range contribution mostly comes from the size/shape parameters of the ions as listed in Table 1. This part is not affected by the substitution method for new interaction parameters, which is intended only for AIOMFAC's middle-range part associated with Eq. 3. We argue that within AIOMFAC's framework, which includes ion–dipole interactions in its middle-range part, such substitutions are reasonable in the absence of sufficient high quality data enabling a direct parameter fit.

4. L438: 'AIOMFAC predictions are substantially lower, by about 0.1 units, than EDB measurements at lower water activity, while they are much better at higher water contents.' Do the authors have any explanation on this observation?

Authors' response: This discrepancy reflects a possible outcome when adopting the group–contribution approach. Since our model is not attempting to only fit these EDB measurements (nor only this system) well, existing discrepancies, e.g. due to an imperfect water–sorbitol interaction representation in the salt-free case or contradicting influence from iodide interactions with other alcohols/polyols (affecting involved functional group interactions with iodide), affect the observed reduced agreement at lower water contents. Furthermore, the relatively poorer model–measurement agreement in this case compared to other systems containing alcohols with iodide salts, means the interaction between sorbitol and iodide is more sensitive to the parameters, particularly at low water contents. A more detailed related discussion of uncertainties is provided in Section 4.2.3.